# DEEP DENOISING: RATE-OPTIMAL RECOVERY OF STRUCTURED SIGNALS WITH A DEEP PRIOR

## ABSTRACT

Deep neural networks provide state-of-the-art performance for image denoising, where the goal is to recover a near noise-free image from a noisy image. The underlying principle is that neural networks trained on large datasets have empirically been shown to be able to generate natural images well from a low-dimensional latent representation of the image. Given such a generator network, or prior, a noisy image can be denoised by finding the closest image in the range of the prior. However, there is little theory to justify this success, let alone to predict the denoising performance as a function of the networks parameters. In this paper we consider the problem of denoising an image from additive Gaussian noise, assuming the image is well described by a deep neural network with ReLu activations functions, mapping a $k$-dimensional latent space to an $n$-dimensional image. We state and analyze a simple gradient-descent-like iterative algorithm that minimizes a non-convex loss function, and provably removes a fraction of $(1 - O(k/n))$ of the noise energy. We also demonstrate in numerical experiments that this denoising performance is, indeed, achieved by generative priors learned from data.

## 1 INTRODUCTION

We consider the image or signal denoising problem, where the goal is to remove noise from an unknown image or signal. In more detail, our goal is to obtain an estimate of an image or signal $y_* \in \mathbb{R}^n$ from

$$y = y_* + \eta,$$

where $\eta$ is unknown noise, often modeled as a zero-mean white Gaussian random variable with covariance matrix $\sigma^2/nI$.

Image denoising relies on modeling or prior assumptions on the image $y_*$. For example, suppose that the image $y_*$ lies in a $k$-dimensional subspace of $\mathbb{R}^n$ denoted by $\mathcal{Y}$. Then we can estimate the original image by finding the closest point in $\ell_2$-distance to the noisy observation $y$ on the subspace $\mathcal{Y}$. The corresponding estimate, denoted by $\hat{y}$, obeys

$$\|\hat{y} - y_*\|^2 \lesssim \sigma^2 \frac{k}{n}, \tag{1}$$

with high probability (throughout, $\|\cdot\|$ denotes the $\ell_2$-norm). Thus, the noise energy is reduced by a factor of $k/n$ over the trivial estimate $\hat{y} = y$ which does not use any prior knowledge of the signal. The denoising rate (1) shows that the more concise the image prior or image representation (i.e., the smaller $k$), the more noise can be removed. If on the other hand the prior (the subspace, in this example) does not include the original image $y_*$, then the error bound (1) increases as we would remove a significant part of the signal along with noise when projecting onto the range of the signal prior. Thus a concise and accurate prior is crucial for denoising.

Real world signals rarely lie in *a priori* known subspaces, and the last few decades of image denoising research have developed sophisticated and accurate image models or priors and algorithms. Examples include models based on sparse representations in overcomplete dictionaries such as wavelets (Donoho, 1995) and curvelets (Starck et al., 2002), and algorithms based on exploiting

self-similarity within images (Dabov et al., 2007). A prominent example of the former class of algorithms is the BM3D (Dabov et al., 2007) algorithm, which achieves state-of-the-art performance for certain denoising problems. However, the nuances of real world images are difficult to describe with handcrafted models. Thus, starting with the paper (Elad & Aharon, 2006) that proposes to learn sparse representation based on training data, it has become common to learn concise representation for denoising (and other inverse problems) from a set of training images.

In 2012, Burger et al. (Burger et al., 2012) applied deep networks to the denoising problem, by training a deep network on a large set of images. Since then, deep learning based denoisers (Zhang et al., 2017) have set the standard for denoising. The success of deep network priors can be attributed to their ability to efficiently represent and learn realistic image priors, for example via auto-decoders (Hinton & Salakhutdinov, 2006) and generative adversarial models (Goodfellow et al., 2014). Over the last few years, the quality of deep priors has significantly improved (Karras et al., 2017; Ulyanov et al., 2017). As this field matures, priors will be developed with even smaller latent code dimensionality and more accurate approximation of natural signal manifolds. Consequently, the representation error from deep priors will decrease, and thereby enable even more powerful denoisers.

As the influence of deep networks in inverse problems grows, it becomes increasingly important to understand their performance at a theoretical level. Given that most optimization approaches for deep learning are first order gradient methods, a justification is needed for why they do not get stuck in local minima. The closest theoretical work to this question is Bora et al. (2017), which solves a noisy compressive sensing problem with generative priors by minimizing empirical risk. Under the assumption that the network is Lipschitz, they show that if the global optimizer can be found, which is in principle NP-hard, then a signal estimate is recovered to within the noise level. While the Lipschitzness assumption is quite mild, the resulting theory does not provide justification for why global optimality can be reached.

The most related work that establishes theoretical reasons for why gradient methods would not get stuck in local minima, when using deep generative priors for solving inverse problems, is Hand & Voroninski (2018). In it, the authors establish global favorability for optimization of the noiseless empirical risk function. Specifically, they show existence of a descent direction outside a ball around the global optimizer and a negative multiple of it in the latent space of the generative model. This work does not provide a specific algorithm which provably estimates the global minimizer, nor does it provide an analysis of the robustness of the problem with respect to noise.

In this paper, we propose the first algorithm for solving denoising with deep generative priors that provably finds an approximation of the underlying image. Following the lead of Hand & Voroninski (2018), we assume an expansive Gaussian model for the deep generative network in order to establish this result.

**Contributions:** The goal of this paper is to analytically quantify the denoising performance of deep-prior based denoisers. Specifically, we characterize the performance of a simple and efficient algorithm for denoising based on a $d$-layer generative neural network $G \colon \mathbb{R}^k \to \mathbb{R}^n$, with $k < n$, and random weights. In more detail, we propose a gradient method with a tweak that attempts to minimize the least-squares loss $f(x) = \frac{1}{2}\|G(x) - y\|^2$ between the noisy image $y$ and an image in the range of the prior, $G(x)$. While $f$ is non-convex, we show that the gradient method yields an estimate $\hat{x}$ obeying

$$\|G(\hat{x}) - y_*\|^2 \lesssim \sigma^2 \frac{k}{n},$$

with high probability, where the notation $\lesssim$ absorbs a constant factor depending on the number of layers of the network, and its expansivity, as discussed in more detail later. Our result shows that the denoising rate of a deep prior based denoiser is determined by the dimension of the latent representation.

We also show in numerical experiments, that this rate—shown to be analytically achieved for random priors—is also experimentally achieved for priors learned from real imaging data.

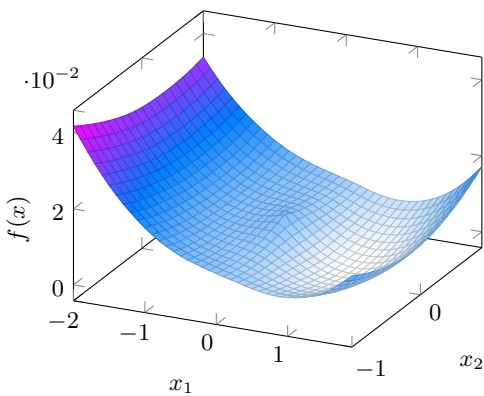

Figure 1: Loss surface $f(x) = \|G(x) - G(x_*)\|$, $x_* = [1,0]$, of an expansive network $G$ with ReLu activation functions with $k = 2$ nodes in the input layer and $n_2 = 300$ and $n_3 = 784$ nodes in the hidden and output layers, respectively, with random Gaussian weights in each layer. The surface has a critical point near $-x_*$, a global minimum at $x_*$, and a local maximum at 0.

## 2   PROBLEM FORMULATION

We consider the problem of estimating a vector $y_* \in \mathbb{R}^n$ from a noisy observation $y = y_* + \eta$. We assume that the vector $y_*$ belongs to the range of a $d$-layer generative neural network $G\colon \mathbb{R}^k \to \mathbb{R}^n$, with $k < n$. That is, $y_* = G(x_*)$ for some $x_* \in \mathbb{R}^k$. We consider a generative network of the form

$$G(x) = \mathrm{relu}(W_d \ldots \mathrm{relu}(W_2 \, \mathrm{relu}(W_1 x_*)) \ldots),$$

where $\mathrm{relu}(x) = \max(x, 0)$ applies entrywise, $W_i \in \mathbb{R}^{n_i \times n_{i-1}}$, are the weights in the $i$-th layer, $n_i$ is the number of neurons in the $i$th layer, and the network is expansive in the sense that $k = n_0 < n_1 < \cdots < n_d = n$. The problem at hand is: Given the weights of the network $W_1 \ldots W_d$ and a noisy observation $y$, obtain an estimate $\hat{y}$ of the original image $y_*$ such that $\|\hat{y} - y_*\|$ is small and $\hat{y}$ is in the range of $G$.

## 3   DENOISING VIA EMPIRICAL RISK MINIMIZATION

As a way to solve the above problem, we first obtain an estimate of $x_*$, denoted by $\hat{x}$, and then estimate $y_*$ as $G(\hat{x})$. In order to estimate $x_*$, we minimize the empirical risk objective

$$f(x) := \frac{1}{2}\|G(x) - y\|^2.$$

Since this objective is nonconvex, there is no *a priori* guarantee of efficiently finding the global minimum. Approaches such as gradient methods could in principle get stuck in local minima, instead of finding a global minimizer that is close to $x_*$.

However, as we show in this paper, under appropriate conditions, a gradient method with a tweak—introduced next—finds a point that is very close to the original latent parameter $x_*$, with the distance to the parameter $x_*$ controlled by the noise. In order to state the algorithm, we first introduce a useful quantity. For analyzing which rows of a matrix $W$ are active when computing $\mathrm{relu}(Wx)$, we let

$$W_{+,x} = \mathrm{diag}(Wx > 0)W.$$

For a fixed weight matrix $W$, the matrix $W_{+,x}$ zeros out the rows of $W$ that do not have a positive dot product with $x$. Alternatively put, $W_{+,x}$ contains weights from only the neurons that are active for the input $x$. We also define $W_{1,+,x} = (W_1)_{+,x} = \mathrm{diag}(W_1 x > 0)W_1$ and

$$W_{i,+,x} = \mathrm{diag}(W_i W_{i-1,+,x} \cdots W_{2,+,x} W_{1,+,x} x > 0)W_i.$$

The matrix $W_{i,+,x}$ consists only of the weights of the neurons in the $i$th layer that are active if the input to the first layer is $x$.

We are now ready to state our algorithm: a gradient method with a tweak informed by the loss surface of the function to be minimized. Given a noisy observation $y$, the algorithm starts with an arbitrary initial point $x_0 \neq 0$. At each iteration $i = 0, 1, \ldots$, the algorithm computes the step direction

$$\tilde{v}_{x_i} = (\Pi_{i=d}^1 W_{i,+,x_i})^t (G(x_i) - y),$$

which is equal to the gradient of $f$ if $f$ is differentiable at $x_i$. It then takes a small step opposite to $\tilde{v}_{x_i}$. The tweak is that before each iteration, the algorithm checks whether $f(-x_i)$ is smaller than $f(x_i)$, and if so, negates the sign of the current iterate $x_i$.

This tweak is informed by the loss surface. To understand this step, it is instructive to examine the loss surface for the *noiseless* case in Figure 1. It can be seen that while the loss function has a *global* minimum at $x_*$, it is relatively flat close to $-x_*$. In expectation, there is a critical point that is a negative multiple of $x_*$ with the property that the curvature in the $\pm x_*$ direction is positive, and the curvature in the orthogonal directions is zero. Further, around approximately $-x_*$, the loss function is larger than around the optimum $x_*$. As a simple gradient descent method (without the tweak) could potentially get stuck in this region, the negation check provides a way to avoid converging to this region. Our algorithm is formally summarized as Algorithm 1 below.

---

**Algorithm 1** Gradient method

---

**Require:** Weights of the network $W_i$, noisy observation $y$, and step size $\alpha > 0$
1: Choose an arbitrary initial point $x_0 \in \mathbb{R}^k \backslash \{0\}$
2: **for** $i = 0, 1, \ldots$ **do**
3:     **if** $f(-x_i) < f(x_i)$ **then**
4:         $x_i \leftarrow -x_i$;
5:     **end if**
6:     Compute $\tilde{v}_{x_i} = (\Pi_{i=d}^1 W_{i,+,x_i})^t (G(x_i) - y)$
7:     $x_{i+1} = x_i - \alpha \tilde{v}_{x_i}$
8: **end for**

---

Other variations of the tweak are also possible. For example, the negation check in Step 2 could be performed after a convergence criterion is satisfied, and if a lower objective is achieved by negating the latent code, then the gradient descent can be continued again until a convergence criterion is again satisfied.

## 4 MAIN RESULTS

For our analysis, we consider a fully-connected generative network $G \colon \mathbb{R}^k \to \mathbb{R}^n$ with Gaussian weights and no bias terms. Specifically, we assume that the weights $W_i$ are independently and identically distributed as $\mathcal{N}(0, 2/n_i)$, but do not require them to be independent across layers. Moreover, we assume that the network is sufficiently *expansive*:

**Expansivity condition.** *We say that the expansivity condition with constant $\epsilon > 0$ holds if*

$$n_i \geqslant c\epsilon^{-2} \log(1/\epsilon) n_{i-1} \log n_{i-1}, \quad \text{for all } i,$$

*where $c$ is a particular numerical constant.*

In a real-world generative network the weights are learned from training data, and are not drawn from a Gaussian distribution. Nonetheless, the motivation for selecting Gaussian weights for our analysis is as follows:

1. The empirical distribution of weights from deep neural networks often have statistics consistent with Gaussians. AlexNet is a concrete example (Arora et al., 2015).

2. The field of theoretical analysis of recovery guarantees for deep learning is nascent, and Gaussian networks can permit theoretical results because of well developed theories for random matrices.

3. It is not clear which non-Gaussian distribution for weights is superior from the joint perspective of realism and analytical tractability.

4. Truly random nets, such as in the Deep Image Prior (Ulyanov et al., 2017), are increasingly becoming of practical relevance. Thus, theoretical advances on random nets is of independent interest.

We are now ready to state our main result.

**Theorem 1.** *Consider a network with the weights in the $i$-th layer, $W_i \in \mathbb{R}^{n_i \times n_{i-1}}$, i.i.d. $\mathcal{N}(0, 2/n_i)$ distributed, and suppose that the network satisfies the expansivity condition for some $\epsilon \leqslant K/d^{90}$. Also, suppose that the noise variance obeys*

$$\omega \leqslant \frac{\|x_*\| K_1}{d^{16}}, \quad \omega := \sqrt{18\sigma^2 \frac{k}{n} \log(n_1^d n_2^{d-1} \dots n_d)}.$$

*Consider the iterates of Algorithm 1 with stepsize $\alpha = K_4 \frac{1}{d^2}$. Then, there exists a number of steps $N$ upper bounded by*

$$N \leqslant \frac{K_2}{d^4 \epsilon} \frac{f(x_0)}{\|x_*\|}$$

*such that after $N$ steps, the iterates of Algorithm 1 obey*

$$\|x_i - x_*\| \leqslant K_5 d^9 \|x_*\| \sqrt{\epsilon} + K_6 d^6 \omega, \quad \text{for all } i \geqslant N, \tag{2}$$

*with probability at least $1 - 2e^{-2k \log n} - \sum_{i=2}^d 8n_i e^{-K_7 n_{i-2}} - 8n_1 e^{-K_7 \epsilon^2 \log(1/\epsilon)k}$. Here, $K_1, K_2, ..$ are numerical constants, and $x_0$ is the initial point in the optimization.*

The error term in the bound (2) consists of two terms—the first is controlled by $\epsilon$, and the second depends on the noise. The first term is negligible if $\epsilon$ is chosen sufficiently small, but that comes at the expense of the expansivity condition being more stringent. The second term in the bound (2) is more interesting and controls the effect of noise. Specifically, for $\epsilon$ sufficiently small, our result guarantees that after sufficiently many iterations,

$$\|x_i - x_*\|^2 \lesssim \sigma^2 \frac{k}{n},$$

where the notation $\lesssim$ absorbs a factor logarithmic in $n$ and polynomial in $d$. One can show that $G$ is Lipschitz in a region around $x_*$[1],

$$\|G(x_i) - G(x_*)\|^2 \lesssim \sigma^2 \frac{k}{n}.$$

Thus, the theorem guarantees that our algorithm yields the denoising rate of $\sigma^2 k/n$, and, as a consequence, denoising based on a generative deep prior provably reduces the energy of the noise in the original image by a factor of $k/n$. We note that the intention of this paper is to show rate-optimality of recovery with respect to the noise power, the latent code dimensionality, and the signal dimensionality. As a result, no attempt was made to establish optimal bounds with respect to the scaling of constants or to powers of $d$. The bounds provided in the theorem are highly conservative in the constants and dependency on the number of layers, $d$, in order to keep the proof as simple as possible. Numerical experiments shown later reveal that the parameter range for successful denoising are much broader than the constants suggest. As this result is the first of its kind for rigorous analysis of denoising performance by deep generative networks, we anticipate the results can be improved in future research, as has happened for other problems, such as sparsity-based compressed sensing and phase retrieval.

## 4.1 THE WEIGHT DISTRIBUTION CONDITION (WDC)

To prove our main result, we make use of a deterministic condition on $G$, called the Weight Distribution Condition (WDC), and then show that Gaussian $W_i$, as given by the statement of Theorem 1 are such that $W_i/\sqrt{2}$ satisfies the WDC with the appropriate probability for all $i$, provided the expansivity condition holds. Our main result, Theorem 1, continues to hold for any weight matrices such that $W_i/\sqrt{2}$ satisfy the WDC.

---

[1]The proof of Lipschitzness follows from applying the Weight Distribution Condition in Section 4.1.

The condition is on the spatial arrangement of the network weights within each layer. We say that the matrix $W \in \mathbb{R}^{n \times k}$ satisfies the *Weight Distribution Condition* with constant $\epsilon$ if for all nonzero $x, y \in \mathbb{R}^k$,

$$\left\| \sum_{i=1}^{n} 1_{\langle w_i, x \rangle > 0} 1_{\langle w_i, y \rangle > 0} \cdot w_i w_i^t - Q_{x,y} \right\| \leqslant \epsilon, \text{ with } Q_{x,y} = \frac{\pi - \theta_0}{2\pi} I_k + \frac{\sin \theta_0}{2\pi} M_{\hat{x} \leftrightarrow \hat{y}}, \quad (3)$$

where $w_i \in \mathbb{R}^k$ is the $i$th row of $W$; $M_{\hat{x} \leftrightarrow \hat{y}} \in \mathbb{R}^{k \times k}$ is the matrix[2] such that $\hat{x} \mapsto \hat{y}$, $\hat{y} \mapsto \hat{x}$, and $z \mapsto 0$ for all $z \in \text{span}(\{x, y\})^\perp$; $\hat{x} = x / \|x\|_2$ and $\hat{y} = y / \|y\|_2$; $\theta_0 = \angle(x, y)$; and $1_S$ is the indicator function on $S$. The norm in the left hand side of (3) is the spectral norm. Note that an elementary calculation[3] gives that $Q_{x,y} = \mathbb{E}[\sum_{i=1}^{n} 1_{\langle w_i, x \rangle > 0} 1_{\langle w_i, y \rangle > 0} \cdot w_i w_i^t]$ for $w_i \sim \mathcal{N}(0, I_k/n)$. As the rows $w_i$ correspond to the neural network weights of the $i$th neuron in a layer given by $W$, the WDC provides a deterministic property under which the set of neuron weights within the layer given by $W$ are distributed approximately like a Gaussian. The WDC could also be interpreted as a deterministic property under which the neuron weights are distributed approximately like a uniform random variable on a sphere of a particular radius. Note that if $x = y$, $Q_{x,y}$ is an isometry up to a factor of $1/2$.

## 5 APPLICATIONS TO COMPRESSED SENSING

In this section we briefly discuss another important scenario to which our results apply to, namely regularizing inverse problems using deep generative priors. Approaches that regularize inverse problems using deep generative models (Bora et al., 2017) have empirically been shown to improve over sparsity-based approaches, see (Lucas et al., 2018) for a review for applications in imaging, and (Mardani et al., 2017) for an application in Magnetic Resonance Imaging showing a significant performance improvement over conventional methods.

Consider an inverse problem, where the goal is to reconstruct an unknown vector $y_* \in \mathbb{R}^n$ from $m < n$ noisy linear measurements:

$$z = Ay_* + \eta \quad \in \mathbb{R}^m,$$

where $A \in \mathbb{R}^{m \times n}$ is called the measurement matrix and $\eta$ is zero mean Gaussian noise with covariance matrix $\sigma^2 / nI$, as before. As before, assume that $y_*$ lies in the range of a generative prior $G$, i.e., $y_* = G(x_*)$ for some $x_*$. As a way to recover $x_*$, consider minimizing the empirical risk objective $f(x) = \frac{1}{2} \|AG(x) - z\|$, using Algorithm 1, with Step 6 substituted by $\tilde{v}_{x_i} = (A\Pi_{i=d}^1 W_{i,+,x_i})^t (AG(x_i) - y)$, to account for the fact that measurements were taken with the matrix $A$.

Suppose that $A$ is a random projection matrix, for concreteness assume that $A$ has i.i.d. Gaussian entries with variance $1/m$. One could prove an analogous result as Theorem 1, but with $\omega = \sqrt{18\sigma^2 \frac{k}{m} \log(n_1^d n_2^{d-1} \ldots n_d)}$, (note that $n$ has been replaced by $m$). This extension shows that, provided $\epsilon$ is chosen sufficiently small, that our algorithm yields an iterate $x_i$ obeying

$$\|G(x_i) - G(x_*)\|^2 \lesssim \sigma^2 \frac{k}{m},$$

where again $\lesssim$ absorbs factors logarithmic in the $n_i$'s, and polynomial in $d$. Proving this result would be analogous to the proof of Theorem 1, but with the additional assumption that the sensing matrix $A$ acts like an isometry on the union of the ranges of $\Pi_{i=d}^1 W_{i,+,x_i}$, analogous to the proof in (Hand & Voroninski, 2018). This extension of our result shows that Algorithm 1 enables solving inverse problems under noise efficiently, and quantifies the effect of the noise.

---

[2] A formula for $M_{\hat{x} \leftrightarrow \hat{y}}$ is as follows. If $\theta_0 = \angle(\hat{x}, \hat{y}) \in (0, \pi)$ and $R$ is a rotation matrix such that $\hat{x}$ and $\hat{y}$ map to $e_1$ and $\cos \theta_0 \cdot e_1 + \sin \theta_0 \cdot e_2$ respectively, then $M_{\hat{x} \leftrightarrow \hat{y}} = R^t \begin{pmatrix} \cos \theta_0 & \sin \theta_0 & 0 \\ \sin \theta_0 & -\cos \theta_0 & 0 \\ 0 & 0 & 0_{k-2} \end{pmatrix} R$, where $0_{k-2}$ is a $k-2 \times k-2$ matrix of zeros. If $\theta_0 = 0$ or $\pi$, then $M_{\hat{x} \leftrightarrow \hat{y}} = \hat{x}\hat{x}^t$ or $-\hat{x}\hat{x}^t$, respectively.

[3] To do this calculation, take $x = e_1$ and $y = \cos \theta_0 \cdot e_1 + \sin \theta_0 \cdot e_2$ without loss of generality. Then each entry of the matrix can be determined analytically by an integral that factors in polar coordinates.

We hasten to add that the paper (Bora et al., 2017) also derived an error bound for minimizing empirical loss. However, the corresponding result (for example Lemma 4.3) differs in two important aspects to our result. First, the result in (Bora et al., 2017) only makes a statement about the *minimizer* of the empirical loss and does not provide justification that an *algorithm* can efficiently find a point near the global minimizer. As the program is non-convex, and as non-convex optimization is NP-hard in general, the empirical loss could have local minima at which algorithms get stuck. In contrast, the present paper presents a specific practical algorithm and proves that it finds a solution near the global optimizer regardless of initialization. Second, the result in (Bora et al., 2017) considers arbitrary noise $\eta$ and thus can not assert denoising performance. In contrast, we consider a random model for the noise, and show the denoising behavior that the resulting error is no more than $O(k/n)$, as opposed to $\|\eta\|^2 \approx O(1)$, which is what we would get from direct application of the result in (Bora et al., 2017).

## 6    EXPERIMENTAL RESULTS

In this section we provide experimental evidence that corroborates our theoretical claims that denoising with deep priors achieves a denoising rate proportional to $\sigma^2 k/n$. We consider both a synthetic, random prior, as studied theoretically in the paper, as well as a prior learned from data. All our results are reproducible with the code provided in the supplement.

### 6.1    DENOISING WITH A SYNTHETIC PRIOR

We start with a synthetic generative network prior with ReLu-activation functions, and draw its weights independently from a Gaussian distribution. We consider a two-layer network with $n = 1500$ neurons in the output layer, $500$ in the middle layer, and vary the number of input neurons, $k$, and the noise level, $\sigma$. We next present simulations showing that if $k$ is sufficiently small, our algorithm achieves a denoising rate proportional to $\sigma k/n$ as guaranteed by our theory.

Towards this goal, we generate Gaussian inputs $x_*$ to the network and observe the noisy image $y = G(x_*) + \eta$, $\eta \sim \mathcal{N}(0, \sigma^2/nI)$. From the noisy image, we first obtain an estimate $\hat{x}$ of the latent representation by running Algorithm 1 until convergence, and second we obtain an estimate of the image as $\hat{y} = G(\hat{x})$. In the left and middle panel of Figure 3, we depict the normalized mean squared error of the latent representation, $\mathrm{MSE}(\hat{x}, x_*)$, and the mean squared error in the image domain, $\mathrm{MSE}(G(\hat{x}), G(x_*))$, where we defined $\mathrm{MSE}(z, z') = \|z - z'\|^2$. For the left panel, we fix the noise variance to $\sigma^2 = 0.25$, and vary $k$, and for the middle panel we fix $k = 50$ and vary the noise variance. The results show that, if the network is sufficiently expansive, guaranteed by $k$ being sufficiently small, then in the noiseless case ($\sigma^2 = 0$), the latent representation and image are perfectly recovered. In the noisy case, we achieve a MSE proportional to $\sigma^2 k/n$, both in the representation and image domains.

We also observed that for the problem instances considered here, the negation trick in step 3-4 of Algorithm 1 is often not necessary, in that even without that step the algorithm typically converges to the global minimum. Having said this, in general the negation step is necessary, since there exist problem instances that have a local minimum opposite of $x_*$.

### 6.2    DENOISING WITH A LEARNED PRIOR

We next consider a prior learned from data. Technically, for such a prior our theory does not apply since we assume the weights to be chosen at random. However, the numerical results presented in this section show that even for the learned prior we achieve the rate predicted by our theory pertaining to a random prior. Towards this goal, we consider a fully-connected autoencoder parameterized by $k$, consisting of an decoder and encoder with ReLu activation functions and fully connected layers. We choose the number of neurons in the three layers of the encoder as $784, 400, k$, and those of the decoder as $k, 400, 784$. We set $k = 10$ and $k = 20$ to obtain two different autoencoders. We train both autoencoders on the MNIST (Lecun et al., 1998) *training set*.

We then take an image $y_*$ from the MNIST *test set*, add Gaussian noise to it, and denoise it using our method based on the learned decoder-network $G$ for $k = 10$ and $k = 20$. Specifically, we estimate

noise variance → 0    0.3    0.6    0.9    1.2    1.5    1.8    2.1    2.4    2.7

noisy →

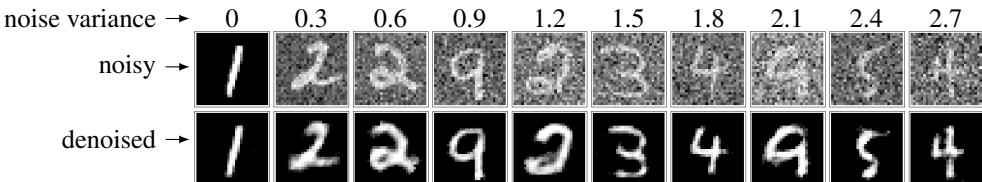

denoised →

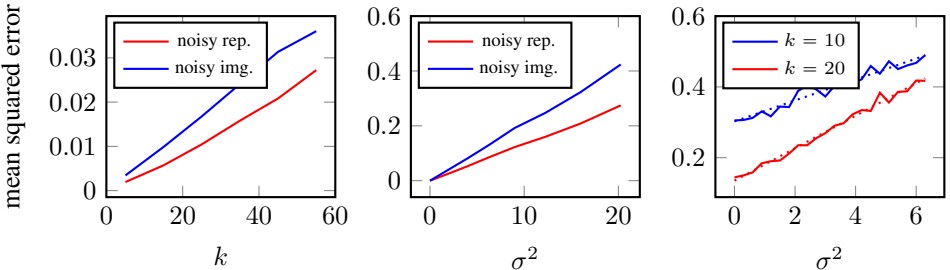

Figure 2: Denosing with a learned generative prior: Even when the number is barely visible, the denoiser recovers a sharp image.

Figure 3: Mean square error in the image domain, $\mathrm{MSE}(G(\hat{x}), x_*)$, and in the latent representation, $\mathrm{MSE}(\hat{x}, x_*)$, as a function of the dimension of the latent representation, $k$, with $\sigma^2 = 0.25$ (**left panel**), and the noise variance, $\sigma^2$ with $k = 50$ (**middle panel**). As suggested by the theory pertaining to decoders with random weights, if $k$ is sufficiently small, and thus the network is sufficiently expansive, the denoising rate is proportional to $\sigma^2 k/n$. **Right panel:** Denoising of handwritten digits based on a learned decoder with $k = 10$ and $k = 20$, along with the least-squares fit as dotted lines. The learned decoder with $k = 20$ has more parameters and thus represents the images with a smaller error; therefore the MSE at $\sigma = 0$ is smaller. However, the denoising rate for the decoder with $k = 20$, which is the slope of the curve is larger as well, as suggested by our theory.

the latent representation $\hat{x}$ by running Algorithm 1, and then set $\hat{y} = G(\hat{x})$. See Figure 2 for a few examples demonstrating the performance of our approach for different noise levels.

We next show that this achieves a mean squared error (MSE) proportional to $\sigma^2 k/n$, as suggested by our theory which applies for decoders with random weights. We add noise to the images with noise variance ranging from $\sigma^2 = 0$ to $\sigma^2 = 6$. In the right panel of Figure 3 we show the MSE in the image domain, $\mathrm{MSE}(G(\hat{x}), G(x_*))$, averaged over a number of images for the learned decoders with $k = 10$ and $k = 20$. We observe an interesting tradeoff: The decoder with $k = 10$ has fewer parameters, and thus does not represent the digits as well, therefore the MSE is larger than that for $k = 20$ for the noiseless case (i.e., for $\sigma = 0$). On the other hand, the smaller number of parameters results in a better denoising rate (by about a factor of two), corresponding to the steeper slope of the MSE as a function of the noise variance, $\sigma^2$.

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

## A PROOFS

In this section we prove our main result, Theorem 1. Instead of proving Theorem 1 as stated, we will prove the following equivalent rescaled statement for when $W_i$ have i.i.d. $\mathcal{N}(0, 1/n_i)$ entries. Because of this rescaling, $G(x)$ scales like $2^{-d/2}\|x\|$, the noise $\omega$ is assumed to scale like $2^{-d/2}$, $\nabla f$ scales like $2^d$, and $\alpha$ scales like $2^d$.

**Theorem 2.** *Consider a network with the weights in the $i$-th layer, $W_i \in \mathbb{R}^{n_i \times n_{i-1}}$, i.i.d. $\mathcal{N}(0, 1/n_i)$ distributed, and suppose that the network satisfies the expansivity condition for some $\epsilon \leqslant K/d^{90}$. Also, suppose that the noise variance obeys*

$$\omega \leqslant \frac{\|x_*\|K_1 2^{-d/2}}{d^{16}}, \quad \omega := \sqrt{18\sigma^2 \frac{k}{n} \log(n_1^d n_2^{d-1} \ldots n_d)}.$$

*Consider the iterates of Algorithm 1 with stepsize $\alpha = K_4 \frac{2^d}{d^2}$. Then, there exists a number of steps $N$ upper bounded by*

$$N \leqslant \frac{K_2}{d^4 \epsilon} \frac{f(x_0)2^d}{\|x_*\|}$$

*such that after $N$ steps, the iterates of Algorithm 1 obey*

$$\|x_i - x_*\| \leqslant K_5 d^9 \|x_*\|\sqrt{\epsilon} + K_6 d^6 2^{d/2} \omega, \quad \text{for all } i \geqslant N, \tag{4}$$

*with probability at least $1 - 2e^{-2k \log n} - \sum_{i=2}^d 8n_i e^{-K_7 n_{i-2}} - 8n_1 e^{-K_7 \epsilon^2 \log(1/\epsilon)k}$. Here, $K_1, K_2, ..$ are numerical constants, and $x_0$ is the initial point in the optimization.*

As mentioned in Section 4.1, our proof makes use of a deterministic condition, called the Weight Distribution Condition (WDC), formally defined in Section 4.1. The following proposition establishes that the expansivity condition ensures that the WDC holds:

**Lemma 3** (Lemma 9 in (Hand & Voroninski, 2018)). *Fix $\epsilon \in (0, 1)$. If the entires of $W_i \in \mathbb{R}^{n_i \times n_{i-1}}$ are i.i.d. $\mathcal{N}(0, 1/n_i)$ and the expansivity condition $n_i > c\epsilon^{-2} \log(1/\epsilon)n_{i-1} \log n_{i-1}$ holds, then $W_i$ satisfies the WDC with constant $\epsilon$ with probability at least $1 - 8n_i e^{-K\epsilon^2 n_{i-1}}$. Here, $c$ and $K$ are numerical constants.*

We note that the form of dependence of $n_i$ on $\epsilon$ can be read off the proofs of Lemma 10 in (Hand & Voroninski, 2018). It follows from Lemma 3, that the WDC holds for all $W_i$ with probability at least $1 - \sum_{i=2}^d 8n_i e^{-K_7 n_{i-2}} - 8n_1 e^{-K_7 \epsilon^2 \log(1/\epsilon)k}$.

In the remainder of the proof we work on the event that the WDC holds for all $W_i$.

### A.1 PRELIMINARIES

Recall that the goal of our algorithm is to minimize the empirical risk objective

$$f(x) = \frac{1}{2}\|G(x) - y\|^2,$$

where $y := G(x_*) + \eta$, with $\eta \sim \mathcal{N}(0, \sigma^2/nI)$.

Our results rely on the fact that outside of two balls around $x = x_*$ and $x = -\rho_d x_*$, with $\rho_d$ a constant defined below, the direction chosen by the algorithm is a descent direction, with high probability. Towards this goal, we use a concentration argument, similar to the arguments used in (Hand & Voroninski, 2018). First, define $\Lambda_x := \Pi_{i=d}^1 W_{i,+,x}$ (with $W_{i,+,x}$ defined in Section 3) for notational convenience, and note that the step direction of our algorithm can be written as

$$\tilde{v}_x = \overline{v}_x + \overline{q}_x, \quad \text{with} \quad \overline{v}_x := \Lambda_x^t \Lambda_x x - (\Lambda_x)^t(\Lambda_{x_*})x_*, \quad \text{and} \quad \overline{q}_x := \Lambda_x^t \eta. \tag{5}$$

Note that at points $x$ where $G$ (and hence $f$) is differentiable, we have that $\tilde{v}_x = \nabla f(x)$.

The proof is based on showing that $\tilde{v}_x$ concentrates around a particular $h_x \in \mathbb{R}^k$, defined below, that is a continuous function of nonzero $x, x_*$ and is zero only at $x = x_*$ and $x = -\rho_d x_*$. The

definition of $h_x$ depends on a function that is helpful for controlling how the operator $x \mapsto W_{+,x}x$ distorts angles, defined as:

$$g(\theta) := \cos^{-1}\Big(\frac{(\pi - \theta)\cos\theta + \sin\theta}{\pi}\Big). \tag{6}$$

With this notation, we define

$$h_x := -\frac{1}{2^d}\Big(\prod_{i=0}^{d-1}\frac{\pi - \overline{\theta}_i}{\pi}\Big)x_* + \frac{1}{2^d}\bigg[x - \sum_{i=0}^{d-1}\frac{\sin\overline{\theta}_i}{\pi}\Big(\prod_{j=i+1}^{d-1}\frac{\pi - \overline{\theta}_j}{\pi}\Big)\frac{\|x_*\|_2}{\|x\|_2}x\bigg],$$

where $\overline{\theta}_0 = \angle(x, x_*)$ and $\overline{\theta}_i = g(\overline{\theta}_{i-1})$. Note that $h_x$ is deterministic and only depends on $x$, $x_*$, and the number of layers, $d$.

In order to bound the deviation of $\tilde{v}_x$ from $h_x$ we use the following two lemmas, bounding the deviation controlled by the WDC and the deviation from the noise:

**Lemma 4** (Lemma 6 in (Hand & Voroninski, 2018)). *Suppose that the WDC holds with $\epsilon < 1/(16\pi d^2)^2$. Then, for all nonzero $x, x_* \in \mathbb{R}^k$,*

$$\|\overline{v}_x - h_x\|_2 \leqslant K\frac{d^3\sqrt{\epsilon}}{2^d}\max(\|x\|_2, \|x_*\|_2), \ and \tag{7}$$

$$\big\langle(\Pi_{i=d}^1 W_{i,+,x})x, (\Pi_{i=d}^1 W_{i,+,x_*})x_*\big\rangle \geqslant \frac{1}{4\pi}\frac{1}{2^d}\|x\|_2\|x_*\|_2, \ and \tag{8}$$

$$\big\|\Pi_{i=d}^1 W_{i,+,x}\big\|^2 \leqslant \frac{1}{2^d}(1 + 2\epsilon)^d \leqslant \frac{13}{12}2^{-d}. \tag{9}$$

*Proof.* Equation (7) and (8) are Lemma 6 in (Hand & Voroninski, 2018). Regarding (9), note that the WDC implies that $\|W_{i,+,x}\|^2 \leqslant 1/2 + \epsilon$. It follows that

$$\big\|\Pi_{i=d}^1 W_{i,+,x}\big\|^2 \leqslant \frac{1}{2^d}(1 + 2\epsilon)^d = \frac{1}{2^d}e^{d\log(1+2\epsilon)} \leqslant \frac{1 + 4\epsilon d}{2^d} \leqslant \frac{13}{12}2^{-d},$$

where the last inequalities follow by our assumption on $\epsilon$. $\qquad\square$

**Lemma 5.** *Suppose the WDC holds with $\epsilon < 1/(16\pi d^2)^2$, that any subset of $n_{i-1}$ rows of $W_i$ are linearly independent for each $i$, and that $\eta \sim \mathcal{N}(0, \sigma^2/nI)$. Then the event*

$$\mathcal{E}_{noise} := \Big\{\big\|(\Pi_{i=d}^1 W_{i,+,x})^t\eta\big\| \leqslant \frac{\omega}{2^{d/2}}, \ for \ all \ x\Big\}, \quad \omega := \sqrt{16\sigma\frac{k}{n}\log(n_1^d n_2^{d-1}\dots n_d)} \tag{10}$$

*holds with probability at least $1 - 2e^{-2k\log n}$.*

As the cost function $f$ is not differentiable everywhere, we will make use of the generalized subdifferential in order to reference the subgradients at nondifferentiable points. For a Lipschitz function $\tilde{f}$ defined from a Hilbert space $\mathcal{X}$ to $\mathbb{R}$, the Clarke generalized directional derivative of $\tilde{f}$ at the point $x \in \mathcal{X}$ in the direction $u$, denoted by $\tilde{f}^o(x; u)$, is defined by $\tilde{f}^o(x; u) = \limsup_{y \to x, t \downarrow 0}\frac{\tilde{f}(y+tu) - \tilde{f}(y)}{t}$, and the generalized subdifferential of $\tilde{f}$ at $x$, denoted by $\partial\tilde{f}(x)$, is defined by

$$\partial\tilde{f}(x) = \{v \in \mathbb{R}^k \mid \langle v, u\rangle \leqslant \tilde{f}^o(x; u), \ for \ all \ u \in \mathcal{X}\}.$$

Since $f(x)$ is a piecewise quadratic function, we have

$$\partial f(x) = \text{conv}(v_1, v_2, \dots, v_t), \tag{11}$$

where conv denotes the convex hull of the vectors $v_1, \dots, v_t$, $t$ is the number of quadratic functions adjoint to $x$, and $v_i$ is the gradient of the $i$-th quadratic function at $x$.

**Lemma 6.** *Under the assumption of Lemma 5, and assuming that $\mathcal{E}_{noise}$ holds, we have that, for any $x \neq 0$ and any $v_x \in \partial f(x)$,*

$$\|v_x - h_x\| \leqslant K\frac{d^3\sqrt{\epsilon}}{2^d}\max(\|x\|_2, \|x_*\|_2) + \frac{\omega}{2^{d/2}}.$$

*In particular, this holds for the subgradient $v_x = \tilde{v}_x$.*

*Proof.* By (11), $\partial f(x) = \text{conv}(v_1, \dots v_t)$ for some finite $t$, and thus $v_x = a_1 v_1 + \dots a_t v_t$ for some $a_1, \dots, a_t \geqslant 0$, $\sum_i a_i = 1$. For each $v_i$, there exists a $w$ such that $v_i = \lim_{t\downarrow 0} \tilde{v}_{x+tw}$. On the event $\mathcal{E}_{\text{noise}}$, we have that for any $x \neq 0$, for any $\tilde{v}_x \in \partial f(x)$

$$
\begin{aligned}
\|\tilde{v}_x - h_x\| &= \|\overline{v}_x + \bar{q}_x - h_x\| \\
&\leqslant \|\overline{v}_x - h_x\| + \|\bar{q}_x\| \\
&\leqslant K \frac{d^3 \sqrt{\epsilon}}{2^d} \max(\|x\|_2, \|x_*\|_2) + \frac{\omega}{2^{d/2}},
\end{aligned}
$$

where the last inequality follows from Lemmas 4 and 5 above. The proof is concluded by appealing to the continuity of $h_x$ with respect to nonzero $x$, and by noting that

$$
\|v_x - h_x\| \leqslant \sum_i a_i \|v_i - h_x\| \leqslant K \frac{d^3 \sqrt{\epsilon}}{2^d} \max(\|x\|_2, \|x_*\|_2) + \frac{\omega}{2^{d/2}},
$$

where we used the inequality above and that $\sum_i a_i = 1$. $\qquad\square$

We will also need an upper bound on the norm of the step direction of our algorithm:

**Lemma 7.** *Suppose that the WDC holds with $\epsilon < 1/(16\pi d^2)^2$ and that the event $\mathcal{E}_{noise}$ holds with $\omega \leqslant \frac{2^{-d/2}\|x_*\|}{8\pi}$. Then, for all $x$,*

$$
\|\tilde{v}_x\| \leqslant \frac{dK}{2^d} \max(\|x\|, \|x_*\|), \tag{12}
$$

*where $K$ is a numerical constant.*

*Proof.* Define for convenience $\zeta_j = \prod_{i=j}^{d-1} \frac{\pi - \bar{\theta}_{j,x,x_*}}{\pi}$. We have

$$
\begin{aligned}
\|\tilde{v}_x\| &\leqslant \|h_x\| + \|h_x - \tilde{v}_x\| \\
&\leqslant \left\| \frac{1}{2^d}x - \frac{1}{2^d}\zeta_0 x_* - \frac{1}{2^d}\sum_{i=0}^{d-1} \frac{\sin\bar{\theta}_{i,x}}{\pi}\zeta_{i+1}\frac{\|x_*\|}{\|x\|}x \right\| + K_1 \frac{d^3\sqrt{\epsilon}}{2^d}\max(\|x\|_2, \|x_*\|_2) + \frac{\omega}{2^{d/2}} \\
&\leqslant \frac{1}{2^d}\|x\| + \left(\frac{1}{2^d} + \frac{d}{\pi 2^d}\right)\|x_*\| + K_1 \frac{d^3\sqrt{\epsilon}}{2^d}\max(\|x\|, \|x_*\|) + \frac{\omega}{2^{d/2}} \\
&\leqslant \frac{dK}{2^d}\max(\|x\|, \|x_*\|),
\end{aligned}
$$

where the second inequality follows from the definition of $h_x$ and Lemma 6, the third inequality uses $|\zeta_j| \leqslant 1$, and the last inequality uses the assumption $\omega \leqslant \frac{2^{-d/2}\|x_*\|}{8\pi}$. $\qquad\square$

## A.2 Proof of Theorem 2

We are now ready to prove Theorem 2. The logic of the proof is illustrated in Figure 4. Recall that $x_i$ is the $i$th iterate of $x$ as per Algorithm 1. We first ensure that we can assume throughout that $x_i$ is bounded away from zero:

**Lemma 8.** *Suppose that WDC holds with $\epsilon < 1/(16\pi d^2)^2$ and that $\mathcal{E}_{noise}$ holds with $\omega$ in (10) obeying $\omega \leqslant \frac{2^{-d/2}\|x_*\|}{8\pi}$. Moreover, suppose that the step size in Algorithm 1 satisfies $0 < \alpha < \frac{K 2^d}{d^2}$, where $K$ is a numerical constant. Then, after at most $N = \left(\frac{38\pi K_0 2^d}{\alpha}\right)^2$ steps, we have that for all $i > N$ that $x_i \notin \mathcal{B}(0, K_0\|x_*\|)$, $K_0 = \frac{1}{32\pi}$.*

In particular, if $\alpha = K 2^d / d^2$, then $N$ is bounded by a constant times $d^4$.

We can therefore assume throughout this proof that $x_i \notin \mathcal{B}(0, K_0\|x_*\|)$, $K_0 = \frac{1}{32\pi}$. We prove Theorem 2 by showing that if $\|h_x\|$ is sufficiently large, i.e., if the iterate $x_i$ is outside of set

$$
\mathcal{S}_\beta = \left\{ x \in \mathbb{R}^k \mid \|h_x\| \leqslant \frac{1}{2^d}\beta \max(\|x\|, \|x_*\|) \right\},
$$

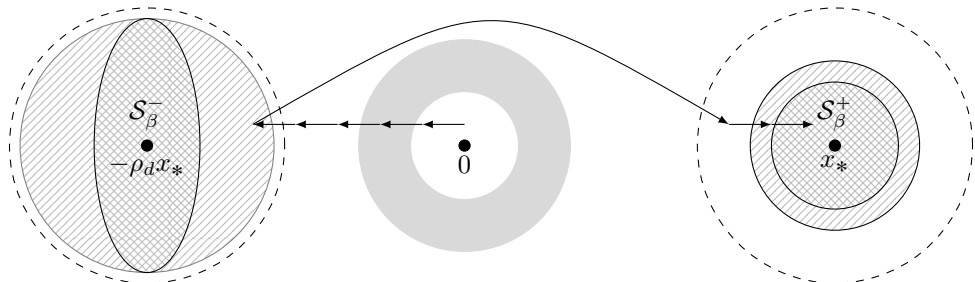

Figure 4: Logic of the proof: Starting at an arbitrary point, Algorithm 1 moves away from $0$, at least till its iterates are outside the gray ring, as $0$ is a local maximum; and once an iterate $x_i$ leaves the gray ring around $0$, all subsequent iterates will never be in the white circle around $0$ again (see Lemma 8). Then the algorithm might move towards $-\rho_d x_*$, but once it enters the dashed ball around $-\rho_d x_*$, it enters a region where the function value is strictly larger than that of the dashed ball around $x_*$, by Lemma 10. Thus steps 3-5 of the algorithm will ensure that the next iterate $x_i$ is in the dashed ball around $x_*$. From there, the iterates will move into the region $\mathcal{S}_\beta^+$, since outside of $\mathcal{S}_\beta^+ \cup \mathcal{S}_\beta^-$ the algorithm chooses a descent direction in each step (see the argument around equation (16)). The region $\mathcal{S}_\beta^+$ is covered by a ball of radius $r$, by Lemma 9, determined by the noise and $\epsilon$.

with

$$\beta = 4Kd^3\sqrt{\epsilon} + 13\omega 2^{d/2}/\|x_*\|, \tag{13}$$

then the algorithm makes progress in the sense that $f(x_{i+1}) - f(x_i)$ is smaller than a certain negative value. The set $\mathcal{S}_\beta$ is contained in two balls around $x_*$ and $-\rho x_*$, whose radius is controlled by $\beta$:

**Lemma 9.** *For any $\beta \leqslant \frac{1}{64^2 d^{12}}$,*

$$\mathcal{S}_\beta \subset \mathcal{B}(x_*, 5000d^6\beta\|x_*\|_2) \cup \mathcal{B}(-\rho_d x_*, 500d^{11}\sqrt{\beta}\|x_*\|_2). \tag{14}$$

*Here, $\rho_d > 0$ is defined in the proof and obeys $\rho_d \to 1$ as $d \to \infty$.*

Note that by the assumption $\omega \leqslant \frac{\|x_*\|K_1 2^{-d/2}}{d^{16}}$ and $Kd^{45}\sqrt{\epsilon} \leqslant 1$, our choice of $\beta$ in (13) obeys $\beta \leqslant \frac{1}{64^2 d^{12}}$ for sufficiently small $K_1, K$, and thus Lemma 9 yields:

$$\mathcal{S}_\beta \subset \mathcal{B}(x_*, r) \cup \mathcal{B}(-\rho_d x_*, \sqrt{r\|x_*\|d^8}).$$

were we define the radius $r = K_2 d^9\sqrt{\epsilon}\|x_*\| + K_3 d^6\omega 2^{d/2}$, where $K_2, K_3$ are numerical constants. Note that hat the radius $r$ is equal to the right hand side in the error bound (4) in our theorem. In order to guarantee that the algorithm converges to a ball around $x_*$, and not to that around $-\rho_d x_*$, we use the following lemma:

**Lemma 10.** *Suppose that the WDC holds with $\epsilon < 1/(16\pi d^2)^2$. Moreover suppose that $\mathcal{E}_{noise}$ holds, and that $\omega$ in the event $\mathcal{E}_{noise}$ obeys $\frac{\omega}{2^{-d/2}\|x_*\|_2} \leqslant K_9/d^2$, where $K_9 < 1$ is a universal constant. Then for any $\phi_d \in [\rho_d, 1]$, it holds that*

$$f(x) < f(y) \tag{15}$$

*for all $x \in \mathcal{B}(\phi_d x_*, K_3 d^{-10}\|x_*\|)$ and $y \in \mathcal{B}(-\phi_d x_*, K_3 d^{-10}\|x_*\|)$, where $K_3 < 1$ is a universal constant.*

In order to apply Lemma 10, define for convenience the two sets:

$$\mathcal{S}_\beta^+ := \mathcal{S}_\beta \cap \mathcal{B}(x_*, r), \text{ and}$$

$$\mathcal{S}_\beta^- := \mathcal{S}_\beta \cap \mathcal{B}(-\rho_d x_*, \sqrt{r\|x_*\|d^8}).$$

By the assumption that $Kd^{45}\sqrt{\epsilon} \leqslant 1$ and $\omega \leqslant K_1 d^{-16} 2^{-d/2}\|x_*\|$, we have that for sufficiently small $K_1, K$,

$$\mathcal{S}_\beta^+ \subseteq \mathcal{B}(x_*, K_3 d^{-10}\|x_*\|) \quad \text{and} \quad \mathcal{S}_\beta^- \subseteq \mathcal{B}(-\rho_d x_*, K_3 d^{-10}\|x_*\|).$$

Thus, the assumptions of Lemma 10 are met, and the lemma implies that for any $x \in \mathcal{S}_\beta^-$ and $y \in \mathcal{S}_\beta^+$, it holds that $f(x) > f(y)$. We now show that the algorithm converges to a point in $\mathcal{S}_\beta^+$. This fact and the negation step in our algorithm (line 3-5) establish that the algorithm converges to a point in $\mathcal{S}_\beta^+$ if we prove that the objective is nonincreasing with iteration number, which will form the remainder of this proof.

Consider $i$ such that $x_i \notin \mathcal{S}_\beta$. By the mean value theorem (Clason, 2017, Theorem 8.13), there is a $t \in [0,1]$ such that for $\hat{x}_i = x_i - t\alpha\tilde{v}_{x_i}$ there is a $v_{\hat{x}_i} \in \partial f(\hat{x}_i)$, where $\partial f$ is the generalized subdifferential of $f$, obeying

$$
\begin{aligned}
f(x_i - \alpha\tilde{v}_{x_i}) - f(x_i) =& \langle v_{\hat{x}_i}, -\alpha\tilde{v}_{x_i}\rangle \\
=& \langle \tilde{v}_{x_i}, -\alpha\tilde{v}_{x_i}\rangle + \langle v_{\hat{x}_i} - \tilde{v}_{x_i}, -\alpha\tilde{v}_{x_i}\rangle \\
\leqslant& -\alpha\|\tilde{v}_{x_i}\|^2 + \alpha\|v_{\hat{x}_i} - \tilde{v}_{x_i}\|\|\tilde{v}_{x_i}\| \\
=& -\alpha\|\tilde{v}_{x_i}\|(\|\tilde{v}_{x_i}\| - \|v_{\hat{x}_i} - \tilde{v}_{x_i}\|).
\end{aligned}
\tag{16}
$$

In the next subsection, we guarantee that for any $t \in [0,1]$, $v_{\hat{x}_i}$ with $\hat{x}_i = x_i - t\alpha\tilde{v}_{x_i}$ is close to $\tilde{v}_{x_i}$:

$$\|v_{\hat{x}_i} - \tilde{v}_{x_i}\| \leqslant \left(\frac{5}{6} + \alpha K_7 \frac{d^2}{2^d}\right)\|\tilde{v}_{x_i}\|, \text{ for all } v_{\hat{x}_i} \in \partial f(\hat{x}_i). \tag{17}$$

Applying (17) to (16) yields

$$f(x_i - \alpha\tilde{v}_{x_i}) - f(x_i) \leqslant -\frac{1}{12}\alpha\|\tilde{v}_{x_i}\|_2^2,$$

where we used that $\alpha K_7 \frac{d^2}{2^d} \leqslant \frac{1}{12}$, by our assumption on the stepsize $\alpha$ being sufficiently small.

Thus, the maximum number of iterations for which $x_i \notin \mathcal{S}_\beta$ is $f(x_0)12/(\alpha \min_i \|\tilde{v}_{x_i}\|^2)$. We next lower-bound $\|\tilde{v}_{x_i}\|$. We have that on $\mathcal{E}_{\text{noise}}$, for all $x \notin \mathcal{S}_\beta$, with $\beta$ given by (13).

$$
\begin{aligned}
\|\tilde{v}_x\|_2 \geqslant& \|h_x\| - \|h_x - \tilde{v}_x\| \\
\geqslant& 2^{-d}\max(\|x\|, \|x_*\|)\left(\beta - K_1 d^3\sqrt{\epsilon} - \omega\frac{2^{d/2}}{\|x_*\|}\right) \\
\geqslant& 2^{-d}\max(\|x\|, \|x_*\|)\left(3Kd^3\sqrt{\epsilon} + 12\omega\frac{2^{d/2}}{\|x_*\|}\right) \\
\geqslant& 2^{-d}\|x_*\|3Kd^3\sqrt{\epsilon}
\end{aligned}
\tag{18}
$$

where the second inequality follows by the definition of $\mathcal{S}_\beta$ and Lemma 6, and the third inequality follows from our definition of $\beta$ in (13). Thus,

$$f(x_i - \alpha\tilde{v}_{x_i}) - f(x_i) \leqslant -\alpha K_5 2^{-2d} d^6 \epsilon\|x_*\|^2 \leqslant -2^{-d} d^4 K_6 \epsilon\|x_*\|^2$$

where we used $\alpha = K_4 \frac{2^d}{d^2}$. Hence, there can be at most $\frac{f(x_0)2^d}{K_6 d^4 \epsilon\|x_*\|^2}$ iterations for which $x_i \notin \mathcal{S}_\beta$.

In order to conclude our proof, we remark that once $x_i$ is inside a ball of radius $r$ around $x_*$, the iterates do not leave a ball of radius $2r$ around $x_*$. To see this, note that by (12) and our choice of stepsize,

$$\alpha\|\tilde{v}_{x_i}\| \leqslant \frac{K}{d}\max(\|x_i\|, \|x_*\|).$$

This concludes our proof.

The remainder of the proof is devoted to prove the lemmas used in this section.

A.3   PROOF OF EQUATION (17)

Our proof relies on $h_x$ being Lipschitz, as formalized by the lemma below, which is proven in Section A.9:

**Lemma 11.** *For any $x, y \notin \mathcal{B}(0, K_0 \|x_*\|)$, where $K_0$ and $K_4$ are numerical constants,*

$$\|h_x - h_y\| \leqslant \frac{K_4 d^2}{2^d} \|x - y\|.$$

By Lemma 11, for all $t \in [0, 1]$ and $i > N$ (recall that by Lemma 8, after at most $N$ steps, $x_i \neq \mathcal{B}(0, K_0 \|x_*\|)$):

$$\|h_{\hat{x}_i} - h_{x_i}\| \leqslant \frac{K_4 d^2}{2^d} \|\hat{x}_i - x_i\|, \tag{19}$$

where $\hat{x}_i = x_i - t\alpha \tilde{v}_{x_i}$. Thus, we have that on $\mathcal{E}_{\text{noise}}$, for any $v_{\hat{x}_i} \in \partial f(\hat{x}_i)$ by Lemma 6,

$$
\begin{aligned}
\|v_{\hat{x}_i} - \tilde{v}_{x_i}\| &\leqslant \|v_{\hat{x}_i} - h_{\hat{x}_i}\| + \|h_{\hat{x}_i} - h_{x_i}\| + \|h_{x_i} - \tilde{v}_{x_i}\| \\
&\leqslant K_1 \frac{d^3 \sqrt{\epsilon}}{2^d} \max(\|\hat{x}_i\|, \|x_*\|) + \frac{\omega}{2^{d/2}} + \frac{K_4 d^2}{2^d} \|\hat{x}_i - x_i\| + K_1 \frac{d^3 \sqrt{\epsilon}}{2^d} \max(\|x_i\|, \|x_*\|) + \frac{\omega}{2^{d/2}} \\
&\leqslant K_1 \frac{d^3 \sqrt{\epsilon}}{2^d} \max(\|x_i\| + \alpha \|\tilde{v}_{x_i}\|, \|x_*\|) + \frac{K_4 d^2}{2^d} \alpha \|\tilde{v}_{x_i}\| + K_1 \frac{d^3 \sqrt{\epsilon}}{2^d} \max(\|x_i\|, \|x_*\|) + 2 \frac{\omega}{2^{d/2}} \\
&\leqslant K_1 \frac{d^3 \sqrt{\epsilon}}{2^d} \left(2 + \frac{\alpha d K}{2^d}\right) \max(\|x_i\|, \|x_*\|) + \frac{K_4 d^2}{2^d} \alpha \|\tilde{v}_{x_i}\| + 2 \frac{K_9/d^2}{2^d} \|x_*\| \tag{20}
\end{aligned}
$$

where the second inequality is from Lemma 6 and Equation 19, and the fourth inequality is from (12) and the assumption $\frac{\omega}{2^{-d/2} \|x_*\|_2} \leqslant K_9/d^2$.

Combining (20) and (18), we get that

$$\|v_{\hat{x}_i} - \tilde{v}_{x_i}\| \leqslant \left(\frac{5}{6} + \alpha K_7 \frac{d^2}{2^d}\right) \|\tilde{v}_{x_i}\|,$$

with the appropriate constants chosen sufficiently small. This concludes the proof of Equation (17).

A.4   PROOF OF LEMMA 8

First suppose that $x_i \in \mathcal{B}(0, 2K_0 \|x_*\|)$. We show that after a polynomial number of iterations $N$, we have that $x_{i+N} \notin \mathcal{B}(0, 2K_0 \|x_*\|)$. Below, we prove that

$$\langle x, \tilde{v}_x \rangle < 0 \text{ and } \|\tilde{v}_x\| \geqslant \frac{1}{2^d 16\pi} \|x_*\| \text{ for all } x \in \mathcal{B}(0, 2K_0 \|x_*\|). \tag{21}$$

It follows that for any $x_i \in \mathcal{B}(0, 2K_0 \|x_*\|)$, $x_i$ and the next iterate produced by the algorithm, $x_{i+1} = x_i - \alpha \tilde{v}_{x_i}$, form an obtruse triangle. As a consequence,

$$
\begin{aligned}
\|x_{i+1}\|^2 &\geqslant \|x_i\|^2 + \alpha^2 \|\tilde{v}_{x_i}\|^2 \\
&\geqslant \|x_i\|^2 + \alpha^2 \frac{1}{(2^d 16\pi)^2} \|x_*\|^2,
\end{aligned}
$$

where the last inequality follows from (21). Thus, the norm of the iterates $x_i$ will increase until after $\left(\frac{2K_0 2^d 16\pi}{\alpha}\right)^2$ iterations, we have $x_{i+N} \notin \mathcal{B}(0, 2K_0 \|x_*\|)$.

The proof of the lemma is concluded by showing that

$$x_i \notin \mathcal{B}(0, 2K_0 \|x_*\|) \text{ implies } x_{i+1} \notin \mathcal{B}(0, K_0 \|x_*\|) \tag{22}$$

As a consequence, in a polynomial number $N$ of steps, for each iterate, we have that $x_i \notin \mathcal{B}(0, K_0 \|x_*\|)$, for all $i \geqslant N$, as claimed.

We next prove the implication (22). Consider $x_i \notin \mathcal{B}(0, 2K_0\|x_*\|)$, and note that

$$
\begin{aligned}
\|x_{i+1}\| = \|x_i - \alpha \tilde{v}_{x_i}\| &\geqslant \|x_i\| - \alpha \|\tilde{v}_{x_i}\| \\
&\geqslant \|x_i\| - \alpha \frac{dK}{2^d} \max(\|x_i\|, \|x_*\|) \\
&\geqslant \|x_i\| - \alpha \frac{dK}{2^d} \frac{\|x_i\|}{2K_0} \\
&\geqslant \|x_i\| - \frac{1}{2}\|x_i\|
\end{aligned}
$$

where the second inequality follows from (12), the third inequality from $\|x_i\| \geqslant 2K_0\|x_*\|$, and finally the last inequality from our assumption on the stepsize $\alpha$. This concludes the proof of (22).

**Proof of (21):** It remains to prove (21). We start with proving $\langle x, \tilde{v}_x \rangle < 0$. For brevity of notation, let $\Lambda_z = \prod_{i=d}^{1} W_{i,+,z}$. We have

$$
\begin{aligned}
x^T \tilde{v}_x = \big\langle \Lambda_x^T \Lambda_x x - \Lambda_x^T \Lambda_{x_*} x_* + \Lambda_x^T \eta, x \big\rangle \\
\leqslant \frac{13}{12} 2^{-d} \|x\|^2 - \frac{1}{4\pi} \frac{1}{2^d} \|x\|\|x_*\| + \|x\| \frac{\omega}{2^{d/2}} \\
\leqslant \|x\| \left( \frac{13}{12} 2^{-d} \|x\| + \frac{1/(8\pi)}{2^d} \|x_*\| - \frac{1}{4\pi} \frac{1}{2^d} \|x_*\| \right) \\
\leqslant \|x\| \frac{1}{2^d} \left( 2\|x\| - \frac{1}{8\pi} \|x_*\| \right).
\end{aligned}
$$

The first inequality follows from (8) and (9), and the second inequality follows from our assumption on $\omega$. Therefore, for any $x \in \mathcal{B}(0, \frac{1}{16\pi}\|x_*\|)$, $\langle x, \tilde{v}_x \rangle < 0$, as desired.

We next show that, for any $x \in \mathcal{B}(0, \frac{1}{16\pi}\|x_*\|)$

$$
\begin{aligned}
\|\tilde{v}_x\| = \|\Lambda_x^T \Lambda_x x - \Lambda_x^T \Lambda_{x_*} x_* + \Lambda_x^T \eta\| &\geqslant \|\Lambda_x^T \Lambda_{x_*} x_*\| - \|\Lambda_x^T \Lambda_x x\| - \|\Lambda_x^T \eta\| \\
&\geqslant \frac{1}{4\pi} \frac{1}{2^d} \|x_*\| - \frac{13}{12} \frac{1}{2^d} \|x\| - \frac{w}{2^{d/2}} \\
&\geqslant \frac{1}{2^d} \left( \frac{1}{8\pi} - \frac{1}{16\pi} \right) \|x_*\|.
\end{aligned}
$$

where the second inequality is from (8) and (9). This concludes the proof of (21).

## A.5 Proof of Lemma 5

Let $\Lambda_x = \Pi_{i=d}^{1} W_{i,+,x}$. We have that

$$
\|\bar{q}_x\|^2 = \|\Lambda_x^t \eta\|^2 \leqslant \|\Lambda_x\|^2 \|P_{\Lambda_x} \eta\|^2,
$$

where $P_{\Lambda_x}$ is a projector onto the span of $\Lambda_x$. As a consequence, $\|P_{\Lambda_x} \eta\|^2$ is $\chi^2$-distributed random variable with $k$-degrees of freedom scaled by $\sigma/n$. A standard tail bound (see (**?**, p. 43)) yields that, for any $\beta \geqslant k$,

$$
\mathbb{P}\left[ \|P_{\Lambda_x} \eta\|^2 \geqslant 4\beta \right] \leqslant 2e^{-\beta}.
$$

Next, we note that by applying Lemmas 13-14 from (Hand & Voroninski, 2018, Proof of Lem. 15))[4], with probability one, that the number of different matrices $\Lambda_x$ can be bounded as

$$
| \{\Lambda_x | x \neq 0\} | = | \{\Pi_{i=d}^{1} W_{i,+,x} | x \neq 0\} | \leqslant 10^{d^2} (n_1^d n_2^{d-1} \ldots n_d)^k \leqslant (n_1^d n_2^{d-1} \ldots n_d)^{2k},
$$

where the second inequality holds for $\log(10) \leqslant k/4 \log(n_1)$. To see this, note that $(n_1^d n_2^{d-1} \ldots n_d)^k \geqslant 10^{d^2}$ is implied by $k(d\log(n_1) + (d-1)\log(n_2) + \ldots \log(n_d)) \geqslant kd^2/4 \log(n_1) \geqslant d^2 \log(10)$. Thus, by the union bound,

$$
\mathbb{P}\left[ \|P_{\Lambda_x} \eta\|^2 \leqslant 16k \log(n_1^d n_2^{d-1} \ldots n_d), \text{ for all } x \right] \geqslant 1 - 2e^{-2k \log(n)},
$$

---

[4]The proof in that argument only uses the assumption of independence of subsets of rows of the weight matrices.

where $n = n_d$. Recall from (9) that $\|\Lambda_x\| \leqslant \frac{13}{12}$. Combining this inequality with $\|\bar{q}_x\|^2 \leqslant \|\Lambda_x\|^2 \|P_{\Lambda_x}\eta\|^2$ concludes the proof.

## A.6    PROOF OF LEMMA 9

We now show that $h_x$ is away from zero outside of a neighborhood of $x_*$ and $-\rho_d x_*$. We prove Lemma 9 by establishing the following:

**Lemma 12.** *Suppose $64d^6\sqrt{\beta} \leqslant 1$. Define*

$$\rho_d := \sum_{i=0}^{d-1} \frac{\sin\breve{\theta}_i}{\pi} \left( \prod_{j=i+1}^{d-1} \frac{\pi - \breve{\theta}_j}{\pi} \right),$$

*where $\breve{\theta}_0 = \pi$ and $\breve{\theta}_i = g(\breve{\theta}_{i-1})$. If $x \in \mathcal{S}_\beta$, then we have that either*

$$|\bar{\theta}_0| \leqslant 32d^4\beta \quad and \quad |\|x\|_2 - \|x_*\|_2| \leqslant 132d^6\beta\|x_*\|_2$$

*or*

$$|\bar{\theta}_0 - \pi| \leqslant 8\pi d^4\sqrt{\beta} \quad and \quad |\|x\|_2 - \|x_*\|_2\rho_d| \leqslant 200d^7\sqrt{\epsilon}\|x_*\|_2.$$

*In particular, we have*

$$\mathcal{S}_\beta \subset \mathcal{B}(x_*, 5000d^6\beta\|x_*\|_2) \cup \mathcal{B}(-\rho_d x_*, 500d^{11}\sqrt{\beta}\|x_*\|_2). \tag{23}$$

*Additionally, $\rho_d \to 1$ as $d \to \infty$.*

*Proof.* Without loss of generality, let $\|x_*\| = 1$, $x_* = e_1$ and $\hat{x} = r\cos\bar{\theta}_0 \cdot e_1 + r\sin\bar{\theta}_0 \cdot e_2$ for $\bar{\theta}_0 \in [0, \pi]$. Let $x \in \mathcal{S}_\beta$.

First we introduce some notation for convenience. Let

$$\xi = \prod_{i=0}^{d-1} \frac{\pi - \bar{\theta}_i}{\pi}, \quad \zeta = \sum_{i=0}^{d-1} \frac{\sin\bar{\theta}_i}{\pi} \prod_{j=i+1}^{d-1} \frac{\pi - \bar{\theta}_j}{\pi}, \quad r = \|x\|_2, \quad M = \max(r, 1).$$

Thus, $h_x = -\frac{1}{2^d}\xi\hat{x}_0 + \frac{1}{2^d}(r - \zeta)\hat{x}$. By inspecting the components of $h_x$, we have that $x \in \mathcal{S}_\beta$ implies

$$|-\xi + \cos\bar{\theta}_0(r - \zeta)| \leqslant \beta M \tag{24}$$

$$|\sin\bar{\theta}_0(r - \zeta)| \leqslant \beta M \tag{25}$$

Now, we record several properties. We have:

$$\bar{\theta}_i \in [0, \pi/2] \text{ for } i \geqslant 1$$

$$\bar{\theta}_i \leqslant \bar{\theta}_{i-1} \text{ for } i \geqslant 1$$

$$|\xi| \leqslant 1 \tag{26}$$

$$|\zeta| \leqslant \frac{d}{\pi}\sin\theta_0 \tag{27}$$

$$\breve{\theta}_i \leqslant \frac{3\pi}{i+3} \text{ for } i \geqslant 0 \tag{28}$$

$$\breve{\theta}_i \geqslant \frac{\pi}{i+1} \text{ for } i \geqslant 0 \tag{29}$$

$$\xi = \prod_{i=0}^{d-1} \frac{\pi - \bar{\theta}_i}{\pi} \geqslant \frac{\pi - \bar{\theta}_0}{\pi}d^{-3} \tag{30}$$

$$\bar{\theta}_0 = \pi + O_1(\delta) \Rightarrow \bar{\theta}_i = \breve{\theta}_i + O_1(i\delta) \tag{31}$$

$$\bar{\theta}_0 = \pi + O_1(\delta) \Rightarrow |\xi| \leqslant \frac{\delta}{\pi} \tag{32}$$

$$\bar{\theta}_0 = \pi + O_1(\delta) \Rightarrow \zeta = \rho_d + O_1(3d^3\delta) \text{ if } \frac{d^2\delta}{\pi} \leqslant 1 \tag{33}$$

We now establish (28). Observe $0 < g(\theta) \leqslant \left(\frac{1}{3\pi} + \frac{1}{\theta}\right)^{-1} =: \tilde{g}(\theta)$ for $\theta \in (0, \pi]$. As $g$ and $\tilde{g}$ are monotonic increasing, we have $\check{\theta}_i = g^{\circ i}(\check{\theta}_0) = g^{\circ i}(\pi) \leqslant \tilde{g}^{\circ i}(\pi) = \left(\frac{i}{3\pi} + \frac{1}{\pi}\right)^{-1} = \frac{3\pi}{i+3}$. Similarly, $g(\theta) \geqslant \left(\frac{1}{\pi} + \frac{1}{\theta}\right)^{-1}$ implies that $\check{\theta}_i \geqslant \frac{\pi}{i+1}$, establishing (29).

We now establish (30). Using (28) and $\overline{\theta}_i \leqslant \check{\theta}_i$, we have

$$\prod_{i=1}^{d-1} \left(1 - \frac{\overline{\theta}_i}{\pi}\right) \geqslant \prod_{i=1}^{d-1} \left(1 - \frac{3}{i+3}\right) \geqslant d^{-3},$$

where the last inequality can be established by showing that the ratio of consecutive terms with respect to $d$ is greater for the product in the middle expression than for $d^{-3}$.

We establish (31) by using the fact that $|g'(\theta)| \leqslant 1$ for all $\theta \in [0, \pi]$ and using the same logic as for (Hand & Voroninski, 2018, Eq. 17).

We now establish (33). As $\overline{\theta}_0 = \pi + O_1(\delta)$, we have $\overline{\theta}_i = \check{\theta}_i + O_1(i\delta)$. Thus, if $\frac{d^2\delta}{\pi} \leqslant 1$,

$$\prod_{j=i+1}^{d-1} \frac{\pi - \overline{\theta}_j}{\pi} = \prod_{j=i+1}^{d-1} \left(\frac{\pi - \check{\theta}_j}{\pi} + O_1\left(\frac{i\delta}{2\pi}\right)\right) = \left(\prod_{j=i+1}^{d-1} \frac{\pi - \check{\theta}_j}{\pi}\right) + O_1(d^2\delta)$$

So

$$\zeta = \sum_{i=0}^{d-1} \left(\frac{\sin\check{\theta}_i}{\pi} + O_1\left(\frac{i\delta}{\pi}\right)\right)\left[\left(\prod_{j=i+1}^{d-1} \frac{\pi - \check{\theta}_j}{\pi}\right) + O_1(d^2\delta)\right] \tag{34}$$

$$= \rho_d + O_1\left(d^2\delta/\pi + d^3\delta/\pi + d^4\delta^2/\pi\right) \tag{35}$$

$$= \rho_d + O_1(3d^3\delta). \tag{36}$$

Thus (33) holds.

Next, we establish that $x \in \mathcal{S}_\beta \Rightarrow r \leqslant 4d$, and thus $M \leqslant 4d$. Suppose $r > 1$. At least one of the following holds: $|\sin\overline{\theta}_0| \geqslant 1/\sqrt{2}$ or $|\cos\overline{\theta}_0| \geqslant 1/\sqrt{2}$. If $|\sin\overline{\theta}_0| \geqslant 1/\sqrt{2}$ then (25) implies that $|r - \zeta| \leqslant \sqrt{2}\beta r$. Using (27), we get $r \leqslant \frac{d/\pi}{1-\sqrt{2}\beta} \leqslant d/2$ if $\beta < 1/4$. If $|\cos\overline{\theta}_0| \geqslant 1/\sqrt{2}$, then (24) implies that $|r - \zeta| \leqslant \sqrt{2}(\beta r + |\xi|)$. Using (26), (27), and $\beta < 1/4$, we get $r \leqslant \frac{\sqrt{2}|\xi|+\zeta}{1-\sqrt{2}\beta} \leqslant \frac{d+\sqrt{2}}{1-\sqrt{2}\beta} \leqslant 4d$. Thus, we have $x \in S_\beta \Rightarrow r \leqslant 4d \Rightarrow M \leqslant 4d$.

Next, we establish that we only need to consider the small angle case ($\overline{\theta}_0 \approx 0$) and the large angle case ($\overline{\theta}_0 \approx \pi$), by considering the following three cases:

(Case I) $\sin\overline{\theta}_0 \leqslant 16d^4\beta$: We have $\overline{\theta}_0 = O_1(32d^4\beta)$ or $\overline{\theta}_0 = \pi + O_1(32d^4\beta)$, as $32d^4\beta < 1$.

(Case II) $|r - \zeta| < \sqrt{\beta}M$: Applying case II to inequality (24) yields $|\xi| \leqslant 2\sqrt{\beta}M$. Using (30), we get $\overline{\theta}_0 = \pi + O_1(2\pi d^3\sqrt{\beta}M)$.

(Case III) $\sin\overline{\theta}_0 > 16d^4\beta$ and $|r - \zeta| \geqslant \sqrt{\beta}M$: Finally, consider Case III. By (25), we have $|r - \zeta| \leqslant \frac{\beta M}{\sin\overline{\theta}_0}$. Using this inequality in (24), we have $|\xi| \leqslant \beta M + \frac{\beta M}{\sin\overline{\theta}_0} \leqslant \frac{2\beta M}{\sin\overline{\theta}_0} \leqslant \frac{1}{8}d^{-4}M \leqslant \frac{1}{2}d^{-3}$, where the second to last inequality uses $\sin\overline{\theta}_0 > 16d^4\beta$ and the last inequality uses $M \leqslant 4d$. By (30), we have $\frac{\pi-\overline{\theta}_0}{\pi}d^{-3} \leqslant \xi \leqslant \frac{1}{2}d^{-3}$, which implies that $\overline{\theta}_0 \geqslant \pi/2$. Now, as $|r - \zeta| \geqslant \sqrt{\beta}M$, then by (25), we have $|\sin\overline{\theta}_0| \leqslant \sqrt{\beta}$. Hence, $\overline{\theta}_0 = \pi + O_1(2\sqrt{\beta})$, as $\overline{\theta}_0 \geqslant \pi/2$ and as $\beta < 1$.

At least one of the Cases I,II, or III hold. Thus, we see that it suffices to consider the small angle case $\overline{\theta}_0 = O_1(32d^4\beta)$ or the large angle case $\overline{\theta}_0 = \pi + O_1(8\pi d^4\sqrt{\beta})$.

**Small Angle Case.** Assume $\overline{\theta}_0 = O_1(\delta)$ with $\delta = 32d^4\beta$. As $\overline{\theta}_i \leqslant \overline{\theta}_0 \leqslant \delta$ for all $i$, we have $1 \geqslant \xi \geqslant (1 - \frac{\delta}{\pi})^d = 1 + O_1(\frac{2\delta d}{\pi})$ provided $\delta d/\pi \leqslant 1/2$ (which holds by our choice $\delta = 32d^4\beta$ by

assumption $64d^6\sqrt{\beta} \leqslant 1$). By (27), we also have $\zeta = O_1(\frac{d}{\pi}\delta)$. By (24), we have

$$|-\xi + \cos\overline{\theta}_0(r - \zeta)| \leqslant \beta M.$$

Thus, as $\cos\overline{\theta}_0 = 1 + O_1(\overline{\theta}_0^2/2) = 1 + O_1(\delta^2/2)$,

$$-\left(1 + O_1(\frac{2\delta d}{\pi})\right) + (1 + O_1(\frac{2\delta d}{\pi}))(r + O_1(\frac{\delta d}{\pi})) = O_1(4d\beta),$$

and $r \leqslant M \leqslant 4d$ (shown above) provides,

$$r - 1 = O_1(4d\beta + \frac{2\delta d}{\pi} + \frac{\delta d}{\pi} + \frac{2\delta d}{\pi}4d + \frac{2\delta^2 d^2}{\pi^2}) \tag{37}$$

$$= O_1(4\beta d + 4\delta d^2). \tag{38}$$

By plugging in that $\delta = 32d^4\beta$, we have that $r - 1 = O_1(132d^6\beta)$, where we have used that $\frac{32d^5\beta}{\pi} \leqslant 1/2$.

**Large Angle Case.** Assume $\theta_0 = \pi + O_1(\delta)$ where $\delta = 8\pi d^4\sqrt{\beta}$. By (32) and (33), we have $\xi = O_1(\delta/\pi)$, and we have $\zeta = \rho_d + O_1(3d^3\delta)$ if $8d^6\sqrt{\beta} \leqslant 1$. By (24), we have

$$|-\xi + \cos\theta_0(r - \zeta)| \leqslant \beta M,$$

so, as $\cos\theta_0 = 1 - O_1(\theta_0^2/2)$,

$$O_1(\delta/\pi) + (1 + O_1(\delta^2/2))(r - \rho_d + O_1(3d^3\delta)) = O_1(\beta M),$$

and thus, using $r \leqslant 4d$, $\rho_d \leqslant d$, and $\delta = 8\pi d^4\sqrt{\beta} \leqslant 1$,

$$r - \rho_d = O_1(\beta M + \delta/\pi + 3d^3\delta + \frac{5}{2}\delta^2 d + \frac{3}{2}d^3\delta^3) \tag{39}$$

$$= O_1\left(4\beta d + \delta(\frac{1}{\pi} + 3d^3 + \frac{5}{2}d + \frac{3}{2}d^3)\right) \tag{40}$$

$$= O_1(200d^7\sqrt{\beta}) \tag{41}$$

To conclude the proof of (23), we use the fact that

$$\|x - x_*\|_2 \leqslant \big|\|x\|_2 - \|x_*\|_2\big| + (\|x_*\|_2 + \big|\|x\|_2 - \|x_*\|_2\big|)\overline{\theta}_0.$$

This fact simply says that if a 2d point is known to have magnitude within $\Delta r$ of some $r$ and is known to be within angle $\Delta\theta$ from 0, then its Euclidean distance to the point of polar coordinates $(r, 0)$ is no more than $\Delta r + (r + \Delta r)\Delta\theta$.

Finally, we establish that $\rho_d \to 1$ as $d \to \infty$. Note that $\rho_{d+1} = (1 - \frac{\breve{\theta}_d}{\pi})\rho_d + \frac{\sin\breve{\theta}_d}{\pi}$ and $\rho_0 = 0$. It suffices to show $\tilde{\rho}_d \to 0$, where $\tilde{\rho}_d := 1 - \rho_d$. The following recurrence relation holds: $\tilde{\rho}_d = (1 - \frac{\breve{\theta}_{d-1}}{\pi})\tilde{\rho}_{d-1} + \frac{\breve{\theta}_{d-1} - \sin\breve{\theta}_{d-1}}{\pi}$, with $\tilde{\rho}_0 = 1$. Using the recurrence formula (Hand & Voroninski, 2018, Eq. (15)) and the fact that $\breve{\theta}_0 = \pi$, we get that

$$\tilde{\rho}_d = \sum_{i=1}^{d} \frac{\breve{\theta}_{i-1} - \sin\breve{\theta}_{i-1}}{\pi} \prod_{j=i+1}^{d} \left(1 - \frac{\breve{\theta}_{j-1}}{\pi}\right) \tag{42}$$

using (29), we have that

$$\prod_{j=i+1}^{d}\left(1 - \frac{\breve{\theta}_{j-1}}{\pi}\right) \leqslant \prod_{j=i+1}^{d}\left(1 - \frac{1}{j}\right) = \exp\left(-\sum_{j=i+1}^{d}\frac{1}{j}\right) \leqslant \exp\left(-\int_{i+1}^{d+1}\frac{1}{s}ds\right) = \frac{i+1}{d+1}$$

Using (28) and the fact that $\breve{\theta}_{i-1} - \sin\breve{\theta}_{i-1} \leqslant \breve{\theta}_{i-1}^3/6$, we have that $\tilde{\rho}_d \leqslant \sum_{i=1}^{d}\frac{\breve{\theta}_{i-1}^3}{6\pi}\cdot\frac{i+1}{d+1} \to 0$ as $d \to \infty$.

$\square$

### A.7   PROOF OF LEMMA 10

Consider the function

$$f_\eta(x) = f_0(x) - \langle G(x) - G(x_*), \eta \rangle,$$

and note that $f(x) = f_\eta(x) + \|\eta\|^2$. Consider $x \in \mathcal{B}(\phi_d x_*, \varphi \|x_*\|)$, for a $\varphi$ that will be specified later. Note that

$$
\begin{aligned}
|\langle G(x) - G(x_*), \eta \rangle| &\leqslant |\langle \Pi_{i=d}^1 W_{i,+,x} x, \eta \rangle| + |\langle \Pi_{i=d}^1 W_{i,+,x_*} x_*, \eta \rangle| \\
&= |\langle x, (\Pi_{i=d}^1 W_{i,+,x})^t \eta \rangle| + |\langle x_*, (\Pi_{i=d}^1 W_{i,+,x_*})^t \eta \rangle| \\
&\leqslant (\|x\| + \|x_*\|) \frac{\omega}{2^{d/2}} \\
&\leqslant (\varphi \|x_*\| + \|x_*\|) \frac{\omega}{2^{d/2}},
\end{aligned}
$$

where the second inequality holds on the event $\mathcal{E}_{\text{noise}}$, by Lemma 5, and the last inequality holds by our assumption on $x$. Thus, for $x \in \mathcal{B}(\phi_d x_*, \varphi \|x_*\|)$

$$
\begin{aligned}
f_\eta(x) \leqslant & \mathbb{E} f_0(x) + |f_0(x) - \mathbb{E} f_0(x)| + |\langle G(x) - G(x_*), \eta \rangle| \\
\leqslant & \frac{1}{2^{d+1}} \left( \phi_d^2 - 2\phi_d + \frac{10}{K_2^3} d\varphi \right) \|x_*\|^2 + \frac{1}{2^{d+1}} \|x_*\|^2 \\
& + \frac{\epsilon(1 + 4\epsilon d)}{2^d} \|x\|^2 + \frac{\epsilon(1 + 4\epsilon d) + 48 d^3 \sqrt{\epsilon}}{2^{d+1}} \|x\| \|x_*\| + \frac{\epsilon(1 + 4\epsilon d)}{2^d} \|x_*\|^2 \\
& + (\varphi \|x_*\| + \|x_*\|) \frac{\omega}{2^{d/2}} \\
\leqslant & \frac{1}{2^{d+1}} \left( \phi_d^2 - 2\phi_d + \frac{10}{K_2^3} d\varphi \right) \|x_*\|^2 + \frac{1}{2^{d+1}} \|x_*\|^2 \\
& + \frac{\epsilon(1 + 4\epsilon d)}{2^d} (\phi_d + \varphi)^2 \|x_*\|^2 + \frac{\epsilon(1 + 4\epsilon d) + 48 d^3 \sqrt{\epsilon}}{2^{d+1}} (\phi_d + \varphi) \|x_*\|^2 + \frac{\epsilon(1 + 4\epsilon d)}{2^d} \|x_*\|^2 \\
& + (\varphi \|x_*\| + \|x_*\|) \frac{\omega}{2^{d/2}} \\
\leqslant & \frac{\|x_*\|^2}{2^{d+1}} \left( 1 + \phi_d^2 - 2\phi_d + \frac{10}{K_2^3} d\epsilon + 68 d^2 \sqrt{\epsilon} \right) + (\varphi \|x_*\| + \|x_*\|) \frac{\omega}{2^{d/2}} \qquad (43)
\end{aligned}
$$

where the last inequality follows from $\epsilon < \sqrt{\epsilon}$, $\rho_d \leqslant 1$, $4\epsilon d < 1$, $\varphi < 1$ and assuming $\varphi = \epsilon$.

Similarly, we have that for any $y \in \mathcal{B}(-\phi_d x_*, \varphi \|x_*\|)$

$$
\begin{aligned}
f_\eta(y) \geqslant & \mathbb{E}[f(y)] - |f(y) - \mathbb{E}[f(y)]| - |\langle G(x) - G(x_*), \eta \rangle| \\
\geqslant & \frac{1}{2^{d+1}} \left( \phi_d^2 - 2\phi_d \rho_d - 10 d^3 \varphi \right) \|x_*\|^2 + \frac{1}{2^{d+1}} \|x_*\|^2 \\
& - \left( \frac{\epsilon(1 + 4\epsilon d)}{2^d} \|y\|^2 + \frac{\epsilon(1 + 4\epsilon d) + 48 d^3 \sqrt{\epsilon}}{2^{d+1}} \|y\| \|x_*\| + \frac{\epsilon(1 + 4\epsilon d)}{2^d} \|x_*\|^2 \right) \\
& - (\varphi \|x_*\| + \|x_*\|) \frac{\omega}{2^{d/2}} \\
\geqslant & \frac{\|x_*\|^2}{2^{d+1}} \left( 1 + \phi_d^2 - 2\phi_d \rho_d - 10 d^3 \varphi - 68 d^2 \sqrt{\epsilon} \right) - (\varphi \|x_*\| + \|x_*\|) \frac{\omega}{2^{d/2}} \qquad (44)
\end{aligned}
$$

Using $\epsilon < \sqrt{\epsilon}$, $\rho_d \leqslant 1$, $4\epsilon d < 1$, $\varphi < 1$ and assuming $\varphi = \epsilon$, the right side of (43) is smaller than the right side of (44) if

$$\varphi = \epsilon \leqslant \left( \frac{\phi_d - \rho_d \phi_d - 13 \|\bar{\eta}\|_2}{\left( 125 + \frac{5}{K_2^3} \right) d^3} \right)^2. \qquad (45)$$

We can establish that:

**Lemma 13.** *For all $d \geqslant 2$, that*

$$1/\left( K_1(d+2)^2 \right) \leqslant 1 - \rho_d \leqslant 250/(d+1).$$

Thus, it suffices to have $\varphi = \epsilon = \frac{K_3}{d^{10}}$ and $13 \|\bar{\eta}\|_2 \leqslant \frac{K_9}{d^2} \leqslant \frac{1}{2} \frac{K_2}{K_1(d+2)^2}$ for an appropriate universal constant $K_9$, and for an appropriate universal constant $K_3$.

## A.8 PROOF OF LEMMA 13

It holds that

$$\|x - y\| \geqslant 2\sin(\theta_{x,y}/2)\min(\|x\|, \|y\|), \qquad \forall x, y \qquad (46)$$

$$\sin(\theta/2) \geqslant \theta/4, \qquad \forall \theta \in [0, \pi] \qquad (47)$$

$$\frac{d}{d\theta}g(\theta) \in [0, 1] \qquad \forall \theta \in [0, \pi] \qquad (48)$$

$$\log(1 + x) \leqslant x \qquad \forall x \in [-0.5, 1] \qquad (49)$$

$$\log(1 - x) \geqslant -2x \qquad \forall x \in [0, 0.75] \qquad (50)$$

where $\theta_{x,y} = \angle(x, y)$. We recall the results (36), (37), and (50) in (Hand & Voroninski, 2018):

$$\check{\theta}_i \leqslant \frac{3\pi}{i+3} \qquad \text{and} \qquad \check{\theta}_i \geqslant \frac{\pi}{i+1} \qquad \forall i \geqslant 0$$

$$1 - \rho_d = \prod_{i=1}^{d-1} \left(1 - \frac{\check{\theta}_i}{\pi}\right) + \sum_{i=1}^{d-1} \frac{\check{\theta}_i - \sin\check{\theta}_i}{\pi} \prod_{j=i+1}^{d-1} \left(1 - \frac{\check{\theta}_j}{\pi}\right).$$

Therefore, we have for all $0 \leqslant i \leqslant d - 2$,

$$\prod_{j=i+1}^{d-1} \left(1 - \frac{\check{\theta}_j}{\pi}\right) \leqslant \prod_{j=i+1}^{d-1} \left(1 - \frac{1}{j+1}\right) = e^{\sum_{j=i+1}^{d-1} \log\left(1 - \frac{1}{j+1}\right)} \leqslant e^{-\sum_{j=i+1}^{d-1} \frac{1}{j+1}} \leqslant e^{-\int_{i+1}^{d} \frac{1}{s+1}ds} = \frac{i+2}{d+1},$$

$$\prod_{j=i+1}^{d-1} \left(1 - \frac{\check{\theta}_j}{\pi}\right) \geqslant \prod_{j=i+1}^{d-1} \left(1 - \frac{3}{j+3}\right) = e^{\sum_{j=i+1}^{d-1} \log\left(1 - \frac{3}{j+3}\right)} \geqslant e^{-\sum_{j=i+1}^{d-1} \frac{6}{j+3}} \geqslant e^{-\int_{i}^{d-1} \frac{6}{s+3}ds} = \left(\frac{i+3}{d+2}\right)^6,$$

where the second and the fifth inequalities follow from (49) and (50) respectively. Since $\pi^3/(12(i+1)^3) \leqslant \check{\theta}_i^3/12 \leqslant \check{\theta}_i - \sin\check{\theta}_i \leqslant \check{\theta}_i^3/6 \leqslant 27\pi^3/(6(i+3)^3)$, we have that for all $d \geqslant 3$

$$1 - \rho_d \leqslant \frac{2}{d+1} + \sum_{i=1}^{d-1} \frac{27\pi^3}{6(i+3)^3} \frac{i+2}{d+1} \leqslant \frac{2}{d+1} + \frac{3\pi^5}{4(d+1)} \leqslant \frac{250}{d+1}$$

and

$$1 - \rho_d \geqslant \left(\frac{3}{(d+2)}\right)^6 + \sum_{i=1}^{d-1} \frac{\pi^3}{12(i+3)^3} \left(\frac{i+3}{d+2}\right)^6 \geqslant \frac{1}{K_1(d+2)^2},$$

where we use $\sum_{i=4}^{\infty} \frac{1}{i^2} \leqslant \frac{\pi^2}{6}$ and $\sum_{i=1}^{n} i^3 = O(n^4)$.

## A.9 PROOF OF LEMMA 11

To establish Lemma 11, we prove the following:

**Lemma 14.** *For all $x, y \neq 0$,*

$$\|h_x - h_y\| \leqslant \left(\frac{1}{2^d} + \frac{6d + 4d^2}{\pi 2^d} \max\left(\frac{1}{\|x\|}, \frac{1}{\|y\|}\right) \|x_*\|\right) \|x - y\|$$

Lemma 11 follows by noting that if $x, y \notin \mathcal{B}(0, r\|x_*\|)$, then $\|h_x - h_y\| \leqslant \left(\frac{1}{2^d} + \frac{6d+4d^2}{\pi r 2^d}\right) \|x - y\|$.

*Proof of Lemma 14.* For brevity of notation, let $\zeta_{j,z} = \prod_{i=j}^{d-1} \frac{\pi - \bar{\theta}_{i,z}}{\pi}$. Combining (46) and (47) gives $|\bar{\theta}_{0,x} - \bar{\theta}_{0,y}| \leqslant 4\max\left(\frac{1}{\|x\|}, \frac{1}{\|y\|}\right) \|x - y\|$. Inequality (48) implies $|\bar{\theta}_{i,x} - \bar{\theta}_{i,y}| \leqslant |\bar{\theta}_{j,x} - \bar{\theta}_{j,y}|$ for all $i \geqslant j$. It follows that

$$\|h_x - h_y\| \leqslant \frac{1}{2^d}\|x - y\| + \frac{1}{2^d} \underbrace{|\zeta_{0,x} - \zeta_{0,y}|}_{T_1} \|x_*\|$$

$$+ \frac{1}{2^d} \underbrace{\left|\sum_{i=0}^{d-1} \frac{\sin\bar{\theta}_{i,x}}{\pi}\zeta_{i+1,x}\hat{x} - \sum_{i=0}^{d-1} \frac{\sin\bar{\theta}_{i,y}}{\pi}\zeta_{i+1,y}\hat{y}\right|}_{T_2} \|x_*\|. \qquad (51)$$

By Lemma 15, we have

$$T_1 \leqslant \frac{d}{\pi} |\bar{\theta}_{0,x} - \bar{\theta}_{0,y}| \leqslant \frac{4d}{\pi} \max\left(\frac{1}{\|x\|}, \frac{1}{\|y\|}\right) \|x - y\|. \tag{52}$$

Additionally, it holds that

$$T_2 = \left| \sum_{i=0}^{d-1} \frac{\sin\bar{\theta}_{i,x}}{\pi} \zeta_{i+1,x} \hat{x} - \frac{\sin\bar{\theta}_{i,x}}{\pi} \zeta_{i+1,x} \hat{y} + \frac{\sin\bar{\theta}_{i,x}}{\pi} \zeta_{i+1,x} \hat{y} - \sum_{i=0}^{d-1} \frac{\sin\bar{\theta}_{i,y}}{\pi} \zeta_{i+1,y} \hat{y} \right|$$

$$\leqslant \frac{d}{\pi} \|\hat{x} - \hat{y}\| + \underbrace{\left| \sum_{i=0}^{d-1} \frac{\sin\bar{\theta}_{i,x}}{\pi} \zeta_{i+1,x} - \sum_{i=0}^{d-1} \frac{\sin\bar{\theta}_{i,y}}{\pi} \zeta_{i+1,y} \right|}_{T_3}. \tag{53}$$

We have

$$T_3 \leqslant \sum_{i=0}^{d-1} \left[ \left| \frac{\sin\bar{\theta}_{i,x}}{\pi} \zeta_{i+1,x} - \frac{\sin\bar{\theta}_{i,x}}{\pi} \zeta_{i+1,y} \right| + \left| \frac{\sin\bar{\theta}_{i,x}}{\pi} \zeta_{i+1,y} - \frac{\sin\bar{\theta}_{i,y}}{\pi} \zeta_{i+1,y} \right| \right]$$

$$\leqslant \sum_{i=0}^{d-1} \left[ \frac{1}{\pi} \left( \frac{d-i-1}{\pi} |\bar{\theta}_{i-1,x} - \bar{\theta}_{i-1,y}| \right) + \frac{1}{\pi} |\sin\bar{\theta}_{i,x} - \sin\bar{\theta}_{i,y}| \right]$$

$$\leqslant \frac{d^2}{\pi} |\bar{\theta}_{0,x} - \bar{\theta}_{0,y}| \leqslant \frac{4d^2}{\pi} \max\left(\frac{1}{\|x\|}, \frac{1}{\|y\|}\right) \|x - y\|. \tag{54}$$

Using (46) and (47) and noting $\|\hat{x} - \hat{y}\| \leqslant \theta_{x,y}$ yield

$$\|\hat{x} - \hat{y}\| \leqslant \theta_{x,y} \leqslant 2 \max\left(\frac{1}{\|x\|}, \frac{1}{\|y\|}\right) \|x - y\|. \tag{55}$$

Finally, combining (51), (52), (53), (54) and (55) yields the result. $\qquad\square$

**Lemma 15.** *Suppose* $a_i, b_i \in [0, \pi]$ *for* $i = 1, \ldots, k$, *and* $|a_i - b_i| \leqslant |a_j - b_j|, \forall i \geqslant j$. *Then it holds that*

$$\left| \prod_{i=1}^{k} \frac{\pi - a_i}{\pi} - \prod_{i=1}^{k} \frac{\pi - b_i}{\pi} \right| \leqslant \frac{k}{\pi} |a_1 - b_1|.$$

*Proof.* Prove by induction. It is easy to verify that the inequality holds if $k = 1$. Suppose the inequality holds with $k = t - 1$. Then

$$\left| \prod_{i=1}^{t} \frac{\pi - a_i}{\pi} - \prod_{i=1}^{t} \frac{\pi - b_i}{\pi} \right| \leqslant \left| \prod_{i=1}^{t} \frac{\pi - a_i}{\pi} - \frac{\pi - a_t}{\pi} \prod_{i=1}^{t-1} \frac{\pi - b_i}{\pi} \right|$$

$$+ \left| \frac{\pi - a_t}{\pi} \prod_{i=1}^{t-1} \frac{\pi - b_i}{\pi} - \prod_{i=1}^{t} \frac{\pi - b_i}{\pi} \right|$$

$$\leqslant \frac{t-1}{\pi} |a_1 - b_1| + \frac{1}{\pi} |a_t - b_t| \leqslant \frac{t}{\pi} |a_1 - b_1|.$$

$\qquad\square$

