# OpenReview forum: "Deep Denoising: Rate-Optimal Recovery of Structured Signals with a Deep Prior"
_ICLR.cc/2019/Conference_

### Official Review · AnonReviewer1 · 2018-10-26
**Interesting theoretical result but very far from practical applicability**

**Rating:** 6
**Confidence:** 4

**Review:**

The paper studies the standard denoising problem under the assumption that the unknown n-dimensional signal can be written as the output of a known d-layer neural network G mapping k dimensions to n dimensions. The paper specifies an algorithm to perform this denoising and the algorithm is based on a variant of the usual gradient method. Then, under additional assumptions on the neural network G, the paper proves that their algorithm produces a denoised signal that achieves a mean squared accuracy of k/n. Because the input signal has "effective" dimensionality k (as it can be written as G(x) for some k-dimensional x), it is nice that it can be recovered at the accuracy k/n by Gradient Descent despite the complicated nature of G. In this respect, the result is quite interesting. However, the underlying assumptions are too strong in my opinion as described below:

1. It is assumed that the Weights of the neural network G are all Gaussian (and also specific Gaussians with mean zero and variances determined by the layer dimensions). This of course is highly impractical. In practice, these network weights are pre-learned (say based on similar datasets) and there is hardly any reason to believe that they will satisfy the Gaussian assumption.
2. It is assumed that the network is expansive in some sense with an expansivity constant \epsilon. This \epsilon then gets into the accuracy bound which basically means that \epsilon has to be set very small. Unfortunately, this leads to the expansivity condition being quite stringent which will further lead to k being very small (especially if d is large). It is unrealistic to believe that real-world signals will come from a neural network with small k.

Given that there do not seem to be other such results for the accuracy of neural network denoising, the paper might still be considered interesting despite the above shortcomings. However, I believe that the theoretical result has near-zero relevance to a practical neural network denoiser.

Another concern is that the paper seems to borrow quite a lot of ideas from the paper "Global Guarantees for Enforcing Deep Generative Priors by Empirical Risk" by Hand and Voroninski. It will be good if the authors can explain the essential differences between the present paper and this earlier paper.

---

> ### Author Response · Authors · 2018-11-14
> **response**
>
> Gaussian assumption:
> There are several justifications for the theoretical study of Gaussian random networks.  First, some trained networks, such as AlexNet, have been shown to exhibit statistics consistent with Gaussians. Thus, all analysis on random networks is potentially of independent interest.  Second, analysis does have to start somewhere.  Weaker assumptions have been made, for example by Bora et al, but they were only able to conclude that a nonconvex optimization's global optimizer is accurate.  As nonconvex optimization is in general NP-hard, it is not clear why a gradient descent algorithm would not get stuck.  The assumptions on this paper are the weakest the authors know the allow the possibility recovery guarantees.  Weakening these assumptions is a worthwhile task and likely challenging task.
>
> k needs to be very small:
> The empirical experiments demonstrate that the regime of applicability of deep denoising by generative model is significantly larger than what the theorem literally states.  Nonetheless, the theorem statements show that all parameter dependencies are polynomial in nature.  From the perspective of theory, this is a significant development, especially as there is no other result for neural network denoising, and one could have expected the results to have exponential dependencies.  Even with the strong assumptions on expansiveness and Gaussianicity, the proofs of favorable behavior by the neural networks are quite technical.  Extending them to weaker realistic assumption would be a worthwhile task now that the first results on this problem have been established.

---

### Official Review · AnonReviewer5 · 2018-11-08
**Interesting but very artificial.**

**Rating:** 5
**Confidence:** 3

**Review:**

This paper studies the signal denoting problem. The theoretical results are nice, and supported by numerical experiments. I have the following two major concerns:

(1) Using deep neural network as a prior in signal denoising is definitely an important and also challenging problem, only when the neural network is learnt from data. However, this paper assumes that the weight matrices of the neural network prior are i.i.d. Gaussian ensemble and independent on the signal. This assumption is oversimplified, and makes the theoretical results become quite expected and delicate. One can hardly get any insights of the practical signal denoising.

(2) The paper has a significant overlap with HV:COLT18:"Global Guarantees for Enforcing Deep Generative Priors by Empirical Risk". HV:COLT18 consider a RIP-type linear operator, and this paper considers the identity operator, which is actually easier. Dealing with the additive noise is new, but somehow incremental.

~~~~~After Rebuttal~~~~~~

The rebuttal still cannot justified such a random deep prior well. I keep my rating unchanged.

---

> ### Author Response · Authors · 2018-11-14
> **response**
>
> Oversimplified assumptions and the insights of practical signal denoising:
> There are several justifications for the theoretical study of Gaussian random networks.  First, some trained networks, such as AlexNet, have been shown to exhibit statistics consistent with Gaussians. Thus, all analysis on random networks is potentially of independent interest.  Second, analysis does have to start somewhere. Weaker assumptions have been made, for example by Bora et al, but they were only able to conclude that a nonconvex optimization's global optimizer is accurate.  As nonconvex optimization is in general NP-hard, it is not clear why a gradient descent algorithm would not get stuck.  The assumptions on this paper are the weakest the authors know that allow the possibility of recovery guarantees.  Weakening these assumptions is a worthwhile and likely challenging task.
>
> Additive noise is incremental:
> The two differences with Hand and Voroninski are that HV did not specifically propose an algorithm that would converge to a neighborhood of the global optimizer, and that HV did not study stability to noise.  Both of these matters are significant because they required additional technical lemmas and additional nontrivial arguments. It is far from obvious that adding noise to each empirical loss term of the objective does not create local minima throughout the search space. This work shows that this is not the case for this network model.

---

### Official Review · AnonReviewer4 · 2018-11-08
**nice theoretical results but under super strong assumptions**

**Rating:** 6
**Confidence:** 3

**Review:**

The paper analyzes the recovery accuracy of a "tweaked" gradient descent algorithm for imaging denoising and compressive sensing under deep generative priors. In particular, when assuming Gaussian randomness of the network weights and extremely stringent conditions of network sizes, they demonstrate a specific denoising rate of O(k/n), with k and n being the input and output dimension of the generative network. This is seemingly optimal in terms of the dependence on the latent code dimensionality and the signal dimensionality and is the first result of this kind.

Two papers are closely related, but are not sufficiently discussed in the introduction. [Bora et al., 2017] does not require Gaussian randomness of the network weights, but achieves only O(1) error bound assuming the empirical risk minimization problem can be solved to optimality.  [Hand & Voroninski, 2018] showed that under same assumptions as in this paper, the nonconvex empirical risk minimization problem exhibits a nice geometric landscape - no spurious stationary points. This implies that virtually anything reasonable would converge to global optimum. Combing both facts, it is not surprising to arrive at the results in this paper.

While the paper makes some novel theoretical contributions, two concerns stand out. First, there is a lack of intuition or justification of the tweak in gradient descent - flipping the sign of the iterate at times.  The author argued that around approximately -x*, the loss function is larger than around the optimum x* . So simple gradient descent is likely to get stuck in this region, so the negation check is needed. I am not so convinced by the argument. There could be other critical points that are not necessarily in the negative regime of true optimal, right?  So why would this be sufficient or necessary for global convergence? Second, even ignoring the unrealistic Gaussian assumption on the network weights, the theorem requires very narrow regimes for the expansivity condition and the noise variance bound. It's hard to verify whether these conditions can be satisfied at all.

The experiment on denoising with learned prior from MNIST data is interesting, as it suggests that the theoretical assumptions are not necessary in practice to observe the optimal recovery rate. It would be more convincing if more experiments are provided, especially for the compressive sensing application.

---

> ### Author Response · Authors · 2018-11-14
> **response**
>
> Discussion of Bora et al. + Hand & Voroninski in introduction:
> We have significantly increased the discussion of these two papers.  In summary: Bora's paper does not establish that the, in principle NP-hard, optimization problem could be solved to global optimality, and Hand & Voroninski does not provide a specific algorithm or show stability with respect to noise.
>
> Intuition on the tweak:
> If the network is Gaussians and sufficiently expansive, then there are not critical points other than the global optimizer and a single negative multiple of it.  Intuitively, this is because the random network model in expectation looks like Figure 1, which shows a single spurious critical point (aside from the local max at zero).  That is why checking only a single alternative point at each step of the gradient descent is reasonable for the particular random generative model that we considered.  In the case that the network is non-Gaussian or is not sufficiently expansive, there may be local minima elsewhere in the landscape, but analyzing these cases would require building a more involved analysis.
>
> Can the expansivity conditions be satisfied at all?:
> Consider the theorem in the case of fixed d.  If epsilon is smaller than a fixed polynomial in d, then a sufficiently expansive network will have each layer grow in width by eps^(-2)log(1/eps) * n log n, where n is the width of the previous layer. By the probability estimate, one can see that the width of each layer should not grow faster than exponentially at each layer.  For such a network, the optimization need only be run a polynomial in d number of iterations, and it receives a point within the noise level plus poly(d)*sqrt(eps) of the underlying original image.
>
> Relevant regime of expansivity condition:
> While the expansivity regime appears restrictive, it is the weakest assumption under which any efficient algorithm has been proven to approximate an image upon denoising by generative model.  Much weaker assumptions, like Lipschitzness, can lead to some theoretical results, like in Bora et al, but those results do not justify why the nonconvex optimization does not get stuck in local minima.  As in Hand and Voroninski, we assume this stronger model in order to have a reasonable chance at establishing a provably convergent algorithm.  It would indeed be wonderful to extend the theoretical analysis to networks that are less expansive, but it may take significant theoretical advances to do that. For this first result on denoising, we note the significance that all scalings are polynomial instead of exponential in nature, and would also like to point out the the result is optimal in the dependence on the latent parameter, k.
>
> Experiments:  The primary message of this paper is denoising by deep generative priors is provably rate optimal for a random model of generative networks.  The purpose of the numerical experiments is to show that the regime of applicability of the theorem is much broader than the theory demonstrates, and that the conclusions drawn by our analysis also applies to a network that is trained on real data.

---

### Meta-Review · Area_Chair1 · 2018-12-17
**Borderline paper: incremental contribution over recent literature**

**Confidence:** 5
**Recommendation:** Reject

**Metareview:**

The paper analyzes the interesting problem of image denoising with neural networks by imposing simplifying assumptions on the Gaussianity and independence of the prior.  A bound is established from the analysis of (Hand & Voroninksi, 2018) that can be algorithmically achieved through a small tweak to gradient descent.

Unfortunately, the contribution of this paper is incremental given the recent works of (Hand & Voroninksi, 2018) and (Bora et al., 2017); an opinion the reviewers unanimously shared.  Reviewer opinion differed on whether they found the overall contribution to be barely acceptable or simply insufficient.  No reviewer detected a major advance, and there seems to be a question of whether the achievement is significant given the strength of the assumptions required to achieve the modest additions.

After scrutiny, the main theoretical contributions of the paper appear to be a bit overstated.  For example, the bound in Theorem 1 is quite weak: it does not establish convergence to a global minimizer (even under the strong assumptions given), but only that Algorithm 1 eventually remains in a neighborhood of the global minimizer.  It is true that this neighborhood can be made arbitrarily small by increasing the strength of the assumptions made on epsilon and omega, but epsilon remains a constant with respect to iteration count.  The subsequent claim that the algorithm achieves a denoising rate of sigma^2 k/n is not an accurate interpretation of Theorem 1, given that this claim would require require (at the very least) that epsilon can be made arbitrarily small, which it cannot be.  More precision is required in stating supportable conclusions from the given results.

The algorithmic motivation itself is rather weak, in the sense that this paper only provides an anecdotal demonstration that there are no spurious critical points beyond the negation of the global minimizer---the theoretical support for this claim already resides in (Hand & Voroninski, 2018).  The provenance of such a central observation was not made sufficiently clear in the paper nor in the discussion.

An additional quibble about the experimental evaluation is that it does not compare to plain gradient descent (or other baseline optimization techniques), which the authors observe almost always works in the scenario considered.  It seems that the "negation tweak" embedded in Algorithm 1 has no real impact on the experimental results, raising the question of whether the contributions do indeed have any practical import.  The descriptions offered in the current paper suggest that a serious algorithmic advantage has yet to be demonstrated in any real experiment.  The paper requires a far better evaluation of Algorithm 1 in comparison to standard baseline optimizers, to support the case that the proposed algorithmic tweak has practical significance.

This paper remained in a weak borderline position after the review and discussion period.  In the end, this was a very difficult decision to make, but I think the paper would benefit from further strengthening before it can constitute a solid publication.